# Remote sensing of young leaf photosynthetic capacity in tropical and subtropical
# evergreen broadleaved forests
Xueqin Yang[1, 2, 6,#], Qingling Sun[2,#], Liusheng Han[3], Jie Tian[2], Wenping Yuan[4], Liyang Liu[5], Wei Zheng[2], Mei
Wang[2], Yunpeng Wang[1,6], Xiuzhi Chen[2,*]
[1] Guangzhou Institute of Geochemistry, Chinese Academy of Sciences, Guangzhou 510640, China
[2] Guangdong Province Data Center of Terrestrial and Marine Ecosystems Carbon Cycle, School of Atmospheric
Sciences, Sun Yat-sen University, Zhuhai 519082, China;
[3] School of Civil Engineering and Geomatics, Shandong University of Technology, Zibo 255000, China;
[4] College of Urban and Environmental Sciences, School of Urban Planning and Design, Peking University, Beijing
100871, China;
[5] Laboratoire des Sciences du Climat et de l'Environnement, IPSL, CEA-CNRS-UVSQ, Université Paris-Saclay,
91191 Gif sur Yvette, France;
[6] College of Earth and Planetary Sciences, University of Chinese Academy of Sciences, Beijing 101408, China;
[#] These authors contributed equally;
[*] *Correspondance: Xiuzhi Chen (chenxzh73@mail.sysu.edu.cn)*
**Abstract**
Determining the large-scale Rubisco carboxylation maximum rate ($V_{c,max25}$) in relation to leaf age is
essential for evaluating the photosynthetic capacity of canopy leaves in global forests. Young leaves ($\leq$180
days), which exhibit higher $V_{c,max25}$ compared to old leaves (>180 days), are key to controlling the seasonality
of leaf photosynthetic capacity in tropical and subtropical evergreen broadleaved forests (TEFs).
Nevertheless, quantifying the leaf photosynthetic capacity of different age across TEFs remains challenging,
especially when considering continuous temporal variations at continental scales. In this study, we propose a
novel methodology that leverages neighborhood pixels analysis with nonlinear least squares optimization to
derive the $V_{c,max25}$ of the young leaves at 0.25° spatial resolution. This approach utilizes satellite-based solar-
induced chlorophyll fluorescence (SIF) products spanning from 2001 to 2018, which were reconstructed
using both the TROPOMI (Tropospheric Monitoring Instrument) SIF and MODIS reflectance data (RTSIF).
Validations against *in situ* observations demonstrate that the newly developed $V_{c,max25}$ products accurately
capture the seasonality of young leaves in South America and subtropical Asia, with correlation coefficients
of 0.84, 0.66, and 0.95, respectively. The $V_{c,max25}$ of the young leaves simulated from the RTSIF-derived gross
primary production (GPP) is effectively correlated (R>0.51) with that dissolved from the global Orbiting
Carbon Observatory-2 (OCO-2)-based SIF (GOSIF) GPP. Furthermore, the gridded $V_{c,max25}$ dataset for young
leaves successfully detects the green-up regions during the dry seasons in the tropics. Overall, this study
presents the first satellite-based $V_{c,max25}$ dataset specifically targeting photosynthetically efficient young
leaves, providing valuable insights for modeling large-scale photosynthetic dynamics and carbon cycle in
TEFs. Herein, we provide the $V_{c,max25}$ time series derived from RTSIF GPP as the primary dataset,
supplemented by GOSIF-derived and FLUXCOM products. These $V_{c,max25}$ products are available at
https://doi.org/10.5281/zenodo.14807414 (Yang et al., 2025).

**Keywords:** maximum rate of carboxylation ($V_{c,max25}$), leaf age, photosynthesis, tropical and subtropical forest.

## 1. Introduction
The maximum carboxylation rate ($V_{c,max}$) is a critical leaf trait that strongly influences the seasonal

variations in canopy photosynthesis across tropical and subtropical evergreen broadleaved forests (TEFs; Chen et al., 2022a; Wu et al., 2018). This relationship stems from the high correlation between the $V_{c,max}$ and nitrogen-related plant functional traits (Lu et al., 2020; Dechant et al., 2020) including leaf nitrogen and chlorophyll content (Lu et al., 2020). However, leaf nitrogen content varies substantially at a large scale due to the influence of multiple biotic and abiotic factors (Quebbeman and Ramirez, 2016), such as leaf lifespan (Onoda et al., 2017), leaf temperature (Verheijen et al., 2013), light intensity (Hikosaka, 2014) and species (Evans, 1989). Leaf nitrogen content inversion from remote sensing data at a large scale remains challenging (Knyazikhin et al., 2013), hindering the accurate mapping of $V_{c,max}$ at regional to global scales.

The $V_{c,max}$ at 25℃ (hereafter denoted as $V_{c,max25}$) serves as a benchmark in most ecosystem models for simulating various $V_{c,max}$ values at different temperatures. For instance, the Farquhar-von Caemmerer-Berry (FvCB) leaf photosynthetic model, widely adopted for simulating plant photosynthesis across ecosystems (Farquhar et al., 1980; Sun et al., 2015), relies on $V_{c,max25}$ as a key parameter in determining leaf photosynthetic capacity. However, $V_{c,max25}$ varies considerably among tree species, with even 2-3-fold differences observed within the same species (Orndahl et al., 2022). Research on this issue remains limited and inconclusive, largely due to the complex interplay of seasonal constraints such as water availability and light, which affect leaf flushing and shedding processes across different climatic zones (Brando et al., 2010; Yang et al., 2021). Recent advancements have led to the development of two independent satellite remote sensing approaches for estimating of $V_{c,max25}$ at a global scale. The first approach to deriving $V_{c,max25}$ is via leaf chlorophyll content (LCC) (Luo et al., 2019; Lu et al., 2020), as chlorophyll harvests light and provides energy for reactions in the Calvin-Benson-Bassham (CBB) cycle of photosynthesis (Luo et al., 2019). Moreover, chlorophyll harvests light energy and powers reactions in the CBB cycle (Luo et al., 2019), $V_{c,max25}$ exhibits strong coordination with LCC as plants optimize their photosynthetic nitrogen resources (Croft et al., 2020; Xu et al., 2022a; Xu et al., 2022b). This LCC-based method enables reliable $V_{c,max25}$ estimation across various spatiotemporal scales. The second approach estimates $V_{c,max25}$ using solar-induced chlorophyll fluorescence (SIF) (Mohammed et al., 2019), which serves as a robust proxy for global gross primary production (GPP) mapping (Mohammed et al., 2019; Frankenberg et al., 2011). Both LCC- and SIF-derived $V_{c,max25}$ products present distinct advantages and limitations. Notably, multispectral satellite data can retrieve LCC at significantly higher spatial and temporal resolutions than SIF measurements (Chen et al., 2022a). Nevertheless, LCC retrieval from remote sensing data is susceptible to uncertainty in the vegetation structural parameters employed in the derivation (Luo et al., 2019). Converting LCC to $V_{c,max25}$ relies on empirical relationships for different plant functional types (PFTs), introducing substantial uncertainties (Chou et al., 2020; Croft et al., 2017; Houborg et al., 2013; Houborg et al., 2015). In comparison, while SIF directly correlates with photosynthetic rates, most satellite-based SIF products suffer from relatively coarse spatial and temporal resolutions (Liu et al., 2024; Chen et al., 2022a). A recent study has demonstrated that TROPOMI SIF data, characterized by high spatial and temporal resolution, exhibit a linear relationship with GPP and contain robust signals for $V_{c,max25}$ (Chen et al., 2022a). Consequently, TROPOMI SIF has been extensively employed for modeling photosynthesis across various ecosystems (Yang et al., 2023).

TEFs account for 40-50% of the carbon sinks in global forest ecosystems, playing a vital role in the global carbon cycle (Yang et al., 2023; Lu et al., 2021). Despite TEFs maintaining a perennial canopy cover, TEFs exhibit pronounced seasonal variability in photosynthetic activity (Wu et al., 2016). This seasonality is primarily attributed to shifts in canopy leaf age structure (Chen et al., 2021; Chen et al., 2022a), which are predominantly driven by climatic seasonality (Li et al., 2021b; Yang et al., 2021). Recent studies have revealed that young leaves (≤180 days) generally exhibit higher $V_{c,max25}$ than old ones (>180 days), thereby dominating the seasonal dynamics of leaf photosynthetic capacity in TEFs (Locke and Ort, 2014; Wu et al.,

2016). Consequently, accurately mapping seasonality of $V_{c,max25}$ seasonality in young leaves is essential for modeling tropical and subtropical photosynthesis at continental scales. However, current satellite-based approaches face challenges in distinguishing $V_{c,max25}$ across leaf age cohorts, primarily due to the complex interactions between climate drivers and leaf phenology (Jensen et al., 2015; Song et al., 2020). These limitations hinder the seasonal characterization of $V_{c,max25}$ of young leaves. Additionally, Earth system models (ESMs) often struggle to capture the seasonal variations in $V_{c,max25}$ across different leaf age categories (Atkin et al., 2014; Ali et al., 2016). A key unresolved challenge remains the insufficient understanding of how seasonal changes in water and light availability regulate leaf emergence and shedding patterns.

To address the aforementioned gaps in mapping $V_{c,max25}$ of young leaves, we categorized the canopy foliage of TEFs into two distinct leaf age groups: young (≤180 days) and old (>180 days) leaves. We then proposed a novel neighbor-based approach to estimate the maximum carboxylation rate ($V_{c,max25}$) for young leaves cohort by assuming a constant for the older cohort (Yang et al., 2023). This assumption is supported by previous research indicating that $V_{c,max25}$ in old leaves exhibits minimal variation over time (Chen et al.,2019; Albert et al., 2018). This study aims to achieve three key objectives: (1) to develop a global gridded dataset capturing seasonal variability of $V_{c,max25}$ in young leaves across TEFs from 2001 to 2018; (2) to validate the dataset against ground-based measurements and dissolved $V_{c,max25}$ data from GOSIF-derived GPP datasets; (3) to analyze the spatiotemporal patterns of $V_{c,max25}$ in young leaves across TEFs. The resulting $V_{c,max25}$ dataset enhances our understanding of tropical and subtropical phenology by quantifying the photosynthetic seasonality of young leaves. Furthermore, it provides valuable insights for refining tropical phenological models within ESMs.

## 2. Materials and methods
### 2.1 Study area
The studied TEFs were identified by selecting pixels marked as EBF (Evergreen Broadleaf Forest; Sulla-Menashe et al., 2018) on MODIS MCD12C1 land cover maps at 0.05° spatial resolution (see **Fig. 1**). TEFs in South America are the largest tropical rainforests in the world and mainly located at 18°N ~22°S and 40~90°W, followed by TEFs in tropical Africa (12°N~12°S, 2.5~37.5°E). TEFs in tropical Asia are mainly located in the Malay Archipelago, Asian Peninsula and northern Australia (30°N~14°S, 85~155°E).

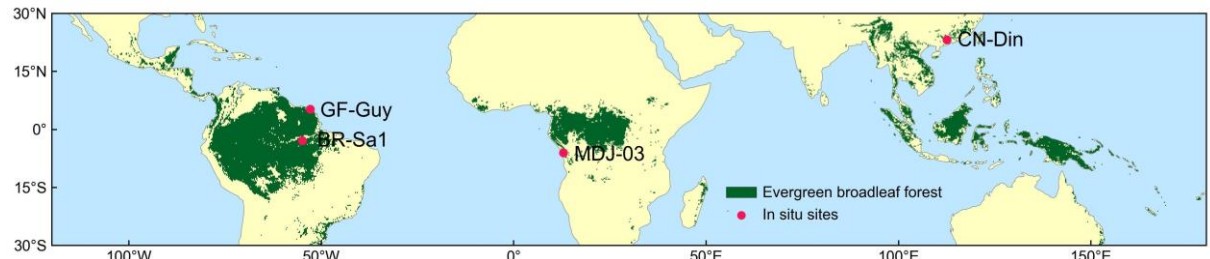

**Figure 1.** Tropical and subtropical evergreen broadleaved forests (TEFs) and *in situ* observation sites. The TEFs is determined as those labeled as evergreen broadleaf forest (EBF) from the MODIS land cover maps at a 0.05° spatial resolution. The red dots are *in situ* observation sites of $V_{c,max25}$.

### 2.2 Data sources for mapping the $V_{c,max25}$ of young leaves
The continental scale GPP (referred to as RTSIF-derived GPP) at a resolution of 0.125° and spanning from 2001 to 2018 was derived from TROPOMI (Tropospheric Monitoring Instrument) SIF data, according to the relationships between the SIF and GPP delineated by Chen et al. (2021), which used a constant value of 15.343 to transform the SIF to the GPP (see **Sect. 2.4.1**). Monthly meteorological data, including the air temperature ($T_{mean}$) from the ERA5-Land dataset (Zhao et al., 2020), vapor pressure deficit (VPD) from

ERA5-Land (Yuan et al., 2019), and downward shortwave solar radiation (SW) provided by the Breathing Earth System Simulator (BESS; Ryu et al., 2018), were used to calculate the Michaelis–Menton constant for carboxylase ($K_C$), the Michaelis–Menton constant for oxygenase ($K_0$), the $CO_2$ compensation point ($\Gamma^*$), dark respiration ($R_d$), and thus to calculate the $A_n$ parameter according to the equations in **Table S1** (see the Supplement). All datasets were collected and harmonized to a spatial resolution of 0.125°. Further details regarding the satellite and input data are provided in **Table 1**.

**Table 1. Data sources for mapping the $V_{c,max25}$ of young leaves across tropical and subtropical evergreen broadleaved forests**

| Data name and Abbr. | Source | Usage | Spatial resolution | Temporal resolution | Temporal coverage |
|---|---|---|---|---|---|
| Temperature ($T_{mean}$) | ERA5-Land | Calculate the $K_C$, $K_0$, $\Gamma^*$, and $R_d$ for $A_n$ | 0.1° × 0.1° | Monthly | 2001.1-2018.12 |
| Shortwave solar radiation (SW) | BESS | Calculate the $J_e$ for $A_n$ | 0.05° × 0.05° | Monthly | 2001.1-2018.12 |
| Vapor pressure deficit (VPD) | ERA5-Land | Calculate the $C_i$ for $A_n$ | 0.1° × 0.1° | Monthly | 2001.1-2018.12 |
| Sun induced chlorophyll fluorescence (RTSIF) | TROPOMI SIF | RTSIF-derived GPP | 0.05° × 0.05° | Monthly | 2001.1-2018.12 |
| Gross primary production retrieved from OCO-2 Solar induced chlorophyll fluorescence (GOSIF) | GOSIF | GOSIF-derived GPP | 0.05° ×0.05° | Monthly | 2001.1-2018.12 |
| Gross primary production from eddy covariance flux tower measurements (FLUXCOM) | FLUXCOM | FLUXCOM GPP | 0.5° × 0.5° | Monthly | 2001.1-2013.12 |
| Leaf-age-dependent leaf area index seasonality product (Lad-LAI) | Yang et al., 2023 | Dissolved $V_{c,max25}$ from GOSIF-derived GPP | 0.25° × 0.25° | Monthly | 2001.1-2018.12 |

## 2.3 Data for validating the $V_{c,max25}$ of young leaves

The $V_{c,max25}$ of the young leaves and canopy-averaged leaves from *in situ* observations were collected to validate the $V_{c,max25}$ seasonality simulated from RTSIF-derived GPP by the proposed model (**Table S2**). Field measurements of monthly $V_{c,max25}$ for young leaves and canopy-averaged leaves were conducted at the Santarem Primary Forest Ecosystem Research Station (BR-Sa1) during August and December 2012 (Albert et al., 2018). Annual $V_{c,max25}$ observations for canopy-averaged leaves were acquired over a 12-months period from 2004 to 2016 at the Guyaflux Forest Ecosystem Research Station (GF-Guy) (Wang et al., 2022), from 2003 to 2009 at the Dinghushan Forest Ecosystem Research Station (CN-Din) (https://fluxnet.org/data/fluxnet2015-dataset/), and in November 2012 at the Mbam–Djerem National Park 3 (MDJ-03) (Ferreira Domingues et al., 2015). The $V_{c,max25}$ of young leaves and canopy-averaged leaves for the BR-Sa1 site were directly obtained from the literature, whereas for the remaining three sites, only existing literature was available, which reported only the $V_{c,max25}$ of canopy-averaged leaves. To evaluate the simulated $V_{c,max25}$ of young leaves, the dissolved method (see **Sect. 2.5.1**) was used to derived the true values of $V_{c,max25}$

for young leaves, based on a monthly leaf-age-dependent leaf area index (Lad-LAI) product (Yang et al.,
2023). Furthermore, gross primary production retrieved from OCO-2 Solar induced chlorophyll fluorescence
(referred to as GOSIF-derived GPP) data at a spatial resolution of 0.05° for the period 2001-2018, and gross
primary production from eddy covariance flux tower measurements (referred to as FLUXCOM GPP) data at
a spatial resolution of 0.5° for the period 2001-2013, were used to evaluate the uncertainty of the proposed
model in simulating monthly gridded $V_{c,max25}$ of young leaves (**Table 1**).

## 2.4 Methods for simulating the $V_{c,max25}$ of young leaves


**Fig. 2** shows the practical procedures applied to produce the seasonal dynamic product of the $V_{c,max25}$ of
young leaves. The 'leaf demographic-identical (LDO)' hypothesis proposes that the leaf cohorts can be
classified into three categories on the basis of their growth, development and lifespan: young leaf (less than
60 days), mature leaf (between 60 days and 180 days), and old leaf (greater than 180 days) (Wu et al., 2017b).
To ensure comparability between the observations and simulations and simplify the calculations, we
categorized the leaf area index (LAI) and the corresponding net $CO_2$ assimilation rate ($A_n$) into two groups
based on leaf age: those with a leaf age greater than 180 days were considered 'old', and those with a leaf
age less than 180 days were considered 'young' (Chen et al., 2020). Since the total GPP of the leaf cohort
remained constant and the leaf cohorts were composed of leaves of different ages, we calculated the total
GPP as a sum of the GPP of each leaf age cohort. The total GPP was simulated using the FvCB photochemical
model by combining the LAI groups (young leaf $LAI_Y$ vs. old leaf $LAI_O$; **Equation 1**) and the corresponding
net assimilation rates of $CO_2$ (young $A_{n,\,sat\_Y}$ vs. old leaf $A_{n,\,sat\_O}$ ; **Equation 1**) (Farquhar et al., 1980).

$$GPP_{total} = LAI_Y \times A_{n,sat\_Y} + LAI_O \times A_{n,sat\_O} \tag{1}$$

where $LAI_Y$ represents the LAI of young leaves (≤180 days) and $LAI_O$ represents the LAI of old leaves (>180
days). $A_{n,\,sat\_Y}$ and $A_{n,\,sat\_O}$ represent the net $CO_2$ assimilation rates of young and old leaves, respectively. The
sum of $LAI_Y$ and $LAI_O$ was set as the total canopy LAI. GPP$_{total}$ refers to the total gross primary production of
the canopy.
The gridded GPP data over the whole TEFs were derived from SIF (denoted as RTSIF-derived GPP)
using a linear SIF–GPP regression model (see **Sect. 2.4.1**), which was established based on *in situ* GPP from
76 eddy covariance (EC) sites (Chen et al., 2022b). The majority of the TEFs retain leaves year-round, and
their total LAI shows marginally small spatial and seasonal changes (Wu et al., 2016; **Fig. S1**). Therefore,
previous modeling studies have assumed a constant value for the total LAI in TEFs (Cramer et al., 2001;
Arora and Boer, 2005; De Weirdt et al., 2012). Based on this, we collected observed seasonal LAI dynamics
in TEFs from previously published literature, which showed a constant value of LAI at around 6.0 (**Fig. S1;**
**Table S3**). Consequently, we streamlined the data to assume that the seasonal LAI was broadly equivalent to
6.0 in TEFs. This assumption was also found to be reasonable in the region of the TEFs by Yang et al. (2023).
The $V_{c,max25}$ values for old leaves were set to 20 µmol m$^{-2}$ s$^{-1}$ according to previous ground-based
observations (Chen et al., 2020; Zhou et al., 2015) in our method. The $A_{n,\,sat\_O}$ can be calculated according to
the FvCB biochemical model (Farquhar et al., 1980; Bernacchi et al., 2003; see **Sect. 2.4.2**). $A_{n,\,sat\_Y}$ can be
expressed as the function of $V_{c,max25}$ for young leaves (see **Sect. 2.4.2**). Consequently, only $LAI_Y$ and $V_{c,max25}$
of young leaves remains as the final parameters to be solved in **Equation 1**.

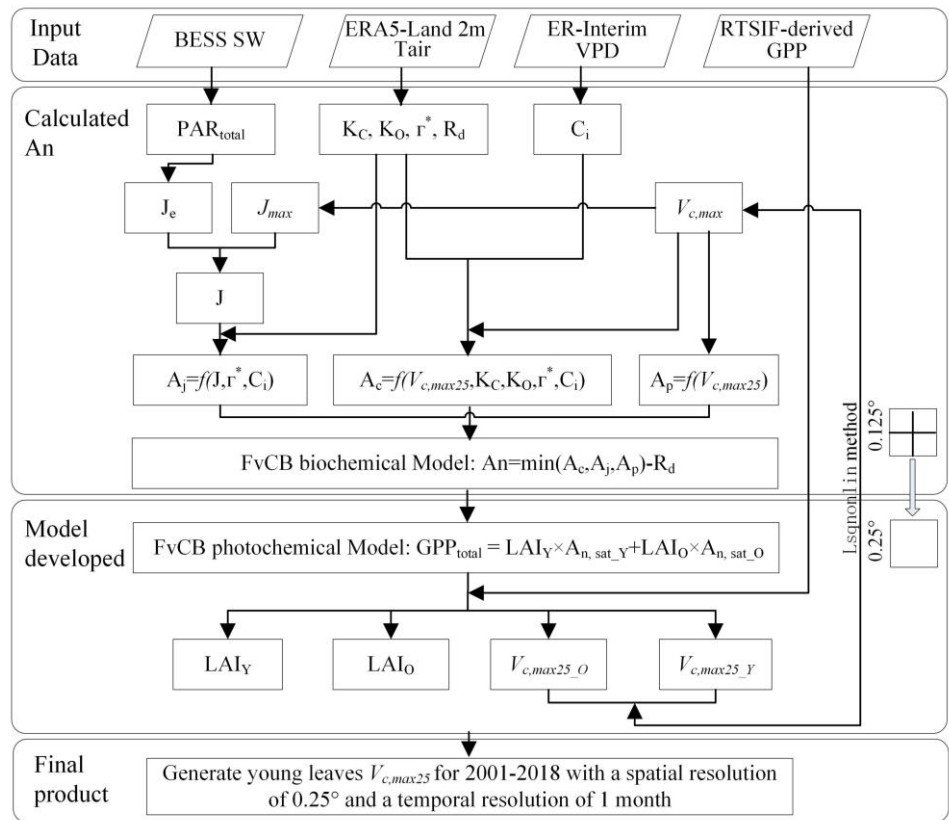

**Figure 2.** Procedures for mapping the $V_{c,max25}$ of young leaves using a neighbor-based approach.

The complexity of model is evident due to the two parameters that needed to be solved. To overcome the challenge of the calculation, we assumed that the four adjacent pixel points had homogeneous plant functional types (PFTs) and had consistent leaf age cohorts. The LAI and $V_{c,max25}$ of young leaves were estimated using nonlinear least squares and constraints on the basis of the GPP values with the four neighboring pixels according to **Equation 1**. The optimal $V_{c,max25}$ was determined by minimizing the residual while satisfying the positivity constraint (i.e., $V_{c,max25} > 0$). The input gridded dataset consisted of the GPP obtained from the RTSIF and climatic data such as $T_{mean}$, VPD and SW. The spatial resolution of these data was homogeneously resampled to 0.125°, resulting in a spatial resolution of 0.25° for the map of the output $V_{c,max25}$ of young leaves. To further validate the robustness and reliability of our neighborhood pixel method, we conducted sensitivity analysis by systematically varying the number of neighborhood pixels, ultimately generating the $V_{c,max25}$ product with 0.5° spatial resolution. In the optimization process, an mean $V_{c,max25}$ value was determined by assuming that the leaf cohort was completely young. A reasonable adjustment for the $V_{c,max25}$ of young leaf value was then determined based on previously published literature (Chen et al., 2021; Yang et al., 2023) and the initial value. Importantly, the difference between the finely optimized $V_{c,max25}$ of young leaves and the initial value could often be significant and outside the margin of error. Therefore, an appropriate adjustment for the $V_{c,max25}$ of young leaves value needs to be carefully determined (He et al., 2019). All analyses were performed using MATLAB (R2 version).

### 2.4.1 Calculating the GPP (RTSIF-derived GPP) from TROPOMI SIF

SIF is a widely used proxy for canopy photosynthesis (Yang et al., 2015; Dechant et al., 2020). Here, we used a long-term reconstructed TROPOMI SIF dataset (RTSIF; Chen et al., 2022b) to estimate GPP seasonality. Previous analyses showed that RTSIF was strongly linearly correlated to eddy covariance (EC)

GPP and used 15.343 as a transformation coefficient to convert RTSIF to GPP (Chen et al., 2022b). We collected seasonal GPP data observed at four EC sites from the FLUXNET2015 tier 1 dataset (**Table S4**; Pastorello et al., 2020) and validated the Chen et al. (2022) simple SIF-GPP relationship (**Fig. S2** in the Supplement). Results confirmed the robustness of the Chen et al. (2022b) simple SIF-GPP relationship for estimating the GPP seasonality in TEFs (R>0.49). Despite the potential overestimation (**Fig. S2a**) or underestimation (**Fig. S2d**) of the magnitudes, the RTSIF-derived GPP mostly captured the seasonality of the EC GPP at all four sites ($d_{phase}$<0.29).

**2.4.2 Calculating the net $CO_2$ assimilation rate**

The net $CO_2$ assimilation rate is a significant parameter characterizing the photosynthetic rate. According to Farquhar's (1980) biochemical model (FvCB), the net $CO_2$ assimilation rate ($A_n$) depends on the most limiting conditions for photosynthesis (RuBisCO saturation $A_c$, RuBP saturation $A_j$, or TPU saturation $A_p$) and the intensity of dark respiration ($R_d$, Bernacchi et al., 2013). The net $CO_2$ assimilation rate (either $A_{n,\,sat\_Y}$ or $A_{n,\,sat\_O}$) can be expressed by the following equation:

$$A_n = \min(A_c, A_j, A_p) - R_d \tag{2}$$

**(1) Calculation of $A_c$**

When the $CO_2$ pressure is low ($C_i$<300 μmol mol$^{-1}$), the net photosynthesis rate is mainly constrained by the activity and quantity of the carboxylase RuBisCO. The Rubisco-limited photosynthetic rate $A_c$ can be calculated using the following equation under a limited carboxylation rate:

$$A_c = V_{cmax} \times \frac{C_i - \Gamma^*}{C_i + K_c \times (1 + \frac{O}{K_0})} \tag{3}$$

where $\Gamma^*$ represents the $CO_2$ compensation point and $C_i$ is the intercellular $CO_2$ pressure. $K_c$, $K_0$, $O$, and $\Gamma^*$ are estimated based on the leaf temperature using **Equation 4** to calculate their values at the given temperature, which is used to convert from their values at 25°.

$$P = P_{25} \times e^{\frac{(T_k - 298.15) \times \Delta H_p}{r \times T_k \times 298.15}} \tag{4}$$

where $P$ is the parameter at each temperature that varies with temperature, including the Michaelis constant for $O_2$ ($K_0$), the Michaelis constant for $CO_2$ ($K_0$), the intercellular concentration ($O$) and the $CO_2$ compensation point ($\Gamma^*$). $P_{25}$ denotes the constant temperature dependence parameter at 25°C (Bernacchi et al., 2001); specifically, $K_c$, $K_0$, $\Gamma^*$ and $O$ at 25°C are equal to 404.9 μmol mol$^{-1}$, 278.4 mmol mol$^{-1}$, 42.75 μmol mol$^{-1}$, and 210 mmol mol$^{-1}$, respectively. $\Delta H_p$ is the activation energy and varies with the temperature and parameters. $r$ is the standard gas constant (8.314 J mol$^{-1}$ K$^{-1}$). $T_k$ is the leaf temperature (unit: K).

Using the stomatal conductance model, the internal $CO_2$ concentration ($C_i$, **Equation 5**) was estimated to depend on the atmospheric $CO_2$ concentration instead of the ambient relative moisture (Xu et al. 2017; Lin et al., 2015; Medlyn et al., 2011).

$$C_i = 380 \times (1 - \frac{1}{1.6 \times (1 + \frac{3.77}{\sqrt{VPD}})}) \tag{5}$$

where $C_i$ represents the internal $CO_2$ concentration.

**(2) Calculation of $A_j$**

When the concentration of $CO_2$ is high, leaf photosynthesis is constrained by RuBP regeneration. The photosynthetic rate ($A_j$) is then limited by electron transport and calculated using the following equation:

$$A_j = J \times \frac{C_i - \Gamma^*}{4 \times (C_i + 2 \times \Gamma^*)} \tag{6}$$

where $J$ is the electron transport rate for leaf photosynthesis. It is a quadratic function of the full electron transfer rate ($J_e$) and maximum electron transfer rate ($J_{max}$) (Luo et al., 2001; Bernacchi et al., 2013). The maximum electron transport rate ($J_{max}$), the maximum carboxylation rate ($V_{c,max25}$), and the $CO_2$ compensation point in the absence of mitochondrial respiration (г*) were used to determine the Michaels–

Menten constants for oxygenation and carboxylase. For the detailed calculation process, refer to **Equations 7-9**.

$$J = \frac{J_e + J_{\max} - \sqrt{(J_e + J_{\max})^2 - 4 \times J_e \times J_{\max} \times \theta}}{2 \times \theta} \quad (7)$$

$$J_e = PAR_{total} \times \partial \times \varphi \times \kappa \quad (8)$$

$$J_{\max} = J_{\max,25} \times e^{\left(\left(\frac{25 - T_{opt}}{\Phi}\right)^2 - \left(\frac{T_k - 273.15 - T_{opt}}{\Phi}\right)^2\right)} \quad (9)$$

where $J_{\max}$ denotes the maximum electron transfer rate at a given temperature and varies with temperature. $J_{\max,25}$ is the maximum electron transfer rate at 25°C, is usually assigned $1.67 \times V_{c,max25}$ in TEFs. $T_{opt}$ is the optimum temperature for electron transfer. $J_e$ is a function of canopy photosynthetically active radiation ($PAR_{total}$) and can be calculated by inputting SW and LAI; for details, refer to Weiss and Norman (1985) and Ryu et al. (2018). $\theta$, $\partial$, $\varphi$, and $\Phi$ are constants and equal to 0.7, 0.85, 0.5, and 0.85, respectively (Xu et al., 2017; Yang et al., 2023). $\kappa$ is a function of the optimal temperature, which represents the maximum quantum efficiency of PSII photochemistry.

**(3) Calculation of $A_p$**

The rate of photosynthesis is limited by the export of triose phosphate. $A_p$ represents the photosynthetic capacity to export or utilize the photosynthetic products for the different LAI cohorts, as determined by multiple field observations.

$$A_p = c \times V_{c,max25} \quad (10)$$

The ratio of the interior foliar $CO_2$ concentration to the environmental $CO_2$ concentration was fixed at 0.5 for $C_3$ species and 0.7 for $C_4$ species based on previous investigations (Fabre et al., 2019; Mcclain and Sharkey, 2019; Yang et al., 2016).

**2.5 Methods for evaluating the simulated $V_{c,max25}$ of young leaves**

This study assessed the proposed algorithms in three ways: (1) monthly *in situ* $V_{c,max25}$ observations obtained from the literature; (2) annually dissolved $V_{c,max25}$ from the GOSIF-derived GPP; and (3) a monthly Lad-LAI product covering the entire TEF region, derived from the RTSIF product by Yang et al. (2023). However, *in situ* $V_{c,max25}$ observations of young leaves remain scarce, with only one site (BR-Sa1; see **Sect. 2.3**) providing monthly $V_{c,max25}$ data. To compensate for the lack of ground-based validation, seasonal $V_{c,max25}$ of canopy-averaged leaves was collected from three additional sites (GF-Guy, MDJ-03 and CN-Din), and the $V_{c,max25}$ of young leaves at these sites was estimated using the dissolved method (see **Sect. 2.5.1**) based on the Lad-LAI product (Yang et al., 2023). To evaluate the efficiency and reliability of the newly proposed methodology, we compared the gridded $V_{c,max25}$ of young leaves simulated from RTSIF-derived GPP using the proposed method with that estimated from GOSIF-derived GPP and the Lad-LAI product using dissolved method. To investigate the reliability of the neighborhood-based subdivision technique, we conducted a comparative analysis for the $V_{c,max25}$ of young leaves derived from RTSIF-derived GPP using 2×2 (0.25° resolution) and 4×4 (0.5° resolution) neighboring pixels. To assess the uncertainties stemming from the estimation of gross primary production, we incorporated two additional GPP products, GOSIF-derived and FLUXCOM GPP (Jung et al., 2019; Yang et al., 2023), along with the original RTSIF-derived GPP, resulting in three distinct versions of the $V_{c,max25}$ of young leaves products.

**2.5.1 Dissolved method for evaluating the $V_{c,max25}$ of young leaves**

The total GPP can be expressed as the sum of the GPP of the young and old cohorts. The GPP of each leaf age cohort is a function of the corresponding LAI cohort and net $CO_2$ assimilation rate. In accordance with related studies, the $V_{c,max25}$ of old leaves is presumed to be a constant value (Chen et al., 2020). When the LAI of different leaf ages is known, only the $V_{c,max25}$ of the young leaves remains unknown in **Equation**

**1**. The value of the $V_{c,max25}$ of the young leaves can be determined by solving the aforementioned **Equation**
**1**. This method involves dividing GPP into young and old cohort according to leaf age, with the $V_{c,max25}$ of
young leaves being directly solved by using the Lad-LAI product, hence the term 'dissolved method'. At
present, there is a lack of available data regarding the ground $V_{c,max25}$ of different leaf ages. The dissolved
method is employed to validate the reasonableness of the proposed algorithm.
**2.5.2 K-means method for classification**
We analyzed the spatial patterns of $V_{c,max25}$ across TEFs using the K-means clustering analysis. K-means
algorithm is an iterative algorithm that tries to partition the dataset into K predefined distinct non-overlapping
subgroups (clusters) where each data point belongs to only one group (Ikotun et al., 2023). It tries to make
the intra-cluster data points as similar as possible while also keeping the clusters as different as possible. It
assigns data points to a cluster such that the sum of the squared distance between the data points and the
cluster's centroid (arithmetic mean of all the data points that belong to that cluster) is at the minimum. Intra-
cluster homogeneity increases as variation decreases, indicating greater similarity among constituent data
points. The way k-means algorithm works is as follows:
(1) Specify number of clusters K.
(2) Initialize centroids by first shuffling the dataset and then randomly selecting K data points for the
centroids without replacement.
(3) Iterate until convergence (i.e., cluster assignments remain unchanged between iterations).
(4) Compute the sum of the squared distance between data points and all centroids.
(5) Assign each data point to the closest cluster (centroid).
(6) Compute the centroids for the clusters by taking the average of all points that belong to each cluster.
**2.5.3 Random forests regression**
Random Forests (RF) is a widely used ensemble learning method that constructs multiple decision trees
through bootstrapped sampling of the training data and aggregates their predictions to enhance model
robustness in regression tasks (Yang et al., 2022). This method is particularly effective in capturing non-
linear relationships and interactions among predictor variables, making it well-suited for complex ecological
datasets. In this study, we employed RF regression to identify the dominant climatic drivers of tropical forest
dynamics across the entire tropical region as well as within three major tropical forest regions. The model
was trained using climate variables as predictors and $V_{c,max25}$ of young leaves as dependent variables. We
utilized the feature importance scores derived from RF to rank the influence of three climatic variables on
forest dynamics across different regions, providing insights into the spatial heterogeneity of climate-forest
interactions.
**2.5.4 Precision evaluation index**
Both the root mean square error (RMSE, **Equation 11**) and Pearson's correlation coefficient (R,
**Equation 12**) were employed to evaluate the model capabilities.
$$RMSE = \sqrt{\frac{\sum_{i=1}^{N}(V_i - U_i)^2}{N}} \tag{11}$$

$$R = \frac{\sum_{i=1}^{N}(V_i - \overline{V})(U_i - \overline{U})}{\sqrt{\sum_{i=1}^{N}(V_i - \overline{V})^2}\sqrt{\sum_{i=1}^{N}(U_i - \overline{U})^2}} \tag{12}$$

$$SD_S = \sqrt{\frac{1}{N}\sum_{i=1}^{n}(V_i - \overline{V})^2} \tag{13}$$

$$SD_m = \sqrt{\frac{1}{N}\sum_{i=1}^{n}(U_i - \overline{U})^2} \tag{14}$$

$$LCS = 2SD_SSD_m(1 - R) \tag{15}$$

where $N$ is the total point extracted from the $V_{c,max25}$ products simulated from RTSIF-derived GPP; $V_i$ and $U_i$ represent the monthly simulated and observed $V_{c,max25}$ *in situ* measurements, respectively; and $\overline{V}$ and $\overline{U}$ are the mean values of the simulated and observed $V_{c,max25}$ *in situ* measurements. Moreover, the continental $V_{c,max25}$ simulated from the proposed model was compared against that the dissolved from GOSIF-derived GPP and Lad-LAI in TEFs. $SD_S$, $SD_m$, and LCS represent the standard deviation of the simulation, standard deviation of the measurement, and the lack of correlation weighted by the standard deviations (phase-related difference; $d_{phase}$), respectively.

**2.6 Quality control (QC) for young leaves $V_{c,max25}$ product**

We provided information on data quality control (QC) along with the $V_{c,max25}$ of young leaves product (**Fig. S3**). In the QC system (**Table S5**), data quality was divided into four levels: Level 1 represents the highest quality, Level 2 and Level 3 represent good and acceptable quality, respectively, and Level 4 should be used with caution. This QC product was generated based on Pearson's correlation coefficients (R) and the root mean square error (RMSE), which were obtained by comparing the seasonal $V_{c,max25}$ estimated from RTSIF- and GOSIF-derived GPP.

## 3. Results

### 3.1 Validation of the gridded $V_{c,max25}$ seasonality of young leaves using *in situ* observations

The seasonality of simulated mean $V_{c,max25}$ for both all canopy leaves and young leaves was evaluated with *in situ* measurements at 4 sites: CN-Din site in southern China (23.17°N, 112.54°E), MDJ-03 site in Congo (5.98°S, 12.87°E), BR-Sa1 (2.86°S, 54.96°W) and GF-Guy (5.28°N, 52.93°W ) sites in southern America. Overall, the estimated mean $V_{c,max25}$ of canopy-averaged leaves (black line) ranged from 20 μmol m$^{-2}$ s$^{-1}$ to 60 μmol m$^{-2}$ s$^{-1}$, and their seasonal fluctuations agreed well with the *in situ* mean $V_{c,max25}$ (red dots) (**Fig. 3**). In contrast, $V_{c,max25}$ of the young leaves (green line) exhibited higher values compared with those of canopy-averaged leaves, ranging from between 40 and 80 μmol m$^{-2}$ s$^{-1}$. This finding is consistent with previous studies that young leaves were more photosynthetically effective than old leaves (Urban et al., 2008; Albert et al., 2018; Menezes et al., 2022). Specifically, our simulations can capture well the seasonal patterns of $V_{c,max25}$ across different sites. At the BR-Sa1 site, the estimates were correlated well with the observed mean $V_{c,max25}$ for all (R=0.85) and young leaves (R=0.84), which both increased during the dry season (approximately between June and December) (**Fig. 3a, 3b**). At the GF-Guy site, *in situ* mean $V_{c,max25}$ of all canopy leaves showed considerable seasonality, while the $V_{c,max25}$ of young leaves remain more stable (**Fig. 3c**). Our estimations also performed well in simulating the $V_{c,max25}$ of all canopy leaves (R=0.95) and that of young leaves (R=0.66) (**Fig. 3d**). In contrast, at the Din site in subtropical Asia, both $V_{c,max25}$ for canopy-averaged leaves and young leaves increased during the wet-season period, with the highest precipitation occurring in June or July (**Fig. 3e**). This is similar in the MDJ-03 site, where both $V_{c,max25}$ for all canopy leaves and young leaves also increased during the wet-season period but with larger seasonal variations. Our model showed the best simulations of $V_{c,max25}$ of young leaves at Din site (canopy-averaged leaves: R=0.84; young leaves: R=0.95). Nevertheless, many more long-term *in situ* measurements are needed to determine the reliability of these time series fluctuations.

Then, we analyzed the spatial patterns of $V_{c,max25}$ across TEFs using the K-means clustering analysis. Results showed that $V_{c,max25}$ for young leaves cohorts had evident seasonal dynamics, bringing influences on canopy photosynthesis. **Fig. S4** shows the timeseries fluctuations in $V_{c,max25}$ for the young leaves in ten

individual regions, as clustered using K-means analysis. Results show the amplitude of $V_{c,max25}$ for young
leaves is smaller in regions closer to the equator and larger in regions farther away from the equator.

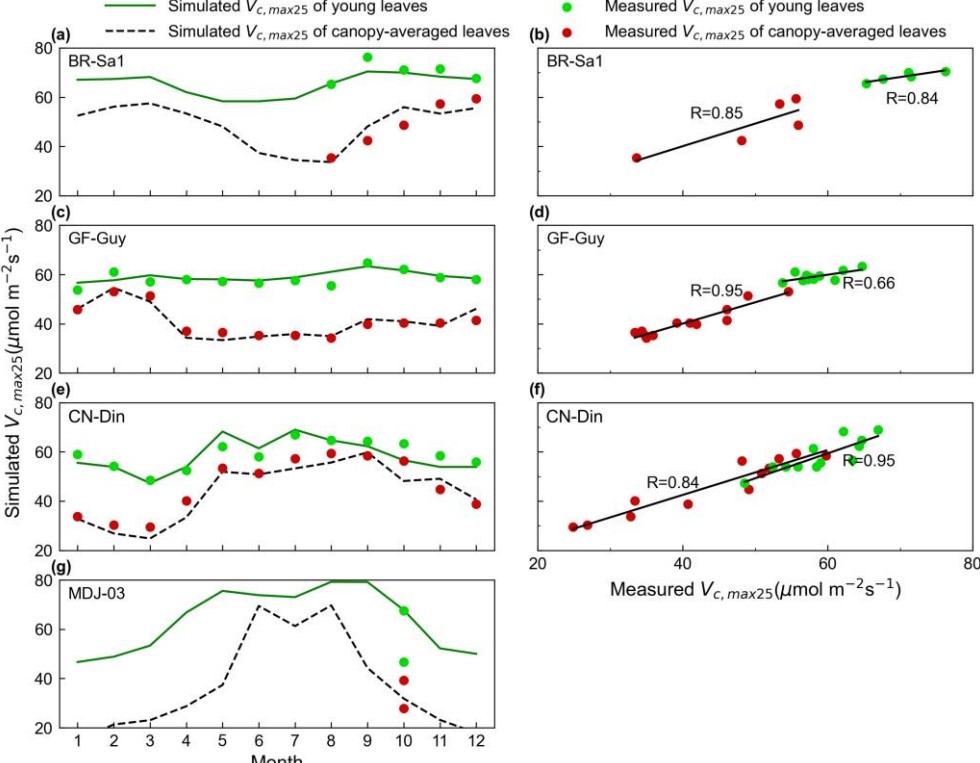

**Figure 3.** Validations of simulated seasonal $V_{c,max25}$ for all canopy leaves and young leaves with *in situ*
observations. The green lines and green dots are the seasonal $V_{c,max25}$ of young leaf simulated from RTSIF
derived GPP and *in situ* observations, respectively. The black dotted line and red dots are the $V_{c,max25}$ of
canopy-averaged leaves from the simulations and *in situ* observations, respectively. Simulated $V_{c,max25}$
denoted as the $V_{c,max25}$ of young leaf simulated from RTSIF-derived GPP by using the new proposed method.


### 3.2 Validation of the $V_{c,max25}$ of young leaves simulated from RTSIF-derived GPP against that dissolved from GOSIF-derived GPP

The $V_{c,max25}$ of young leaves simulated from RTSIF-derived GPP demonstrated significant correlations
(R ranges from 0.51 to 0.87) with those dissolved from GOSIF-derived GPP (**Fig. 4**). Across the Amazon,
more than 69.78% of pixels have a high EBF fraction (>90%). The spatial clustering pattern aligns with the
onset of the dry season (cf. Tang and Dubayah, 2017), suggesting that the clustering analysis effectively
differentiates climate regions within the Amazon. The relatively homogeneous environmental conditions
across these sub-regions create similar plant growth environments, leading to a more constrained range of
$V_{c,max25}$ values and pronounced clustering effects in sub-regions A1–A5. Notably, sub-region A3, located in
the northwestern Amazon near coastal and mountainous areas, forms two distinct clustering zones. Statistical
analysis revealed strong seasonal correlations between the $V_{c,max25}$ of young leaf simulated from RTSIF-
derived and GOSIF-derived GPP, with R>0.80 occupy 91.68% (**Fig. 5a-c**) and RMSE<11.59 occupy 91.68%
(**Fig. 5d-f**) of the TEFs. The K-means spatial clustering analysis revealed strong agreement between the
$V_{c,max25}$ of young leaves simulated by RTSIF-derived and GOSIF-derived GPP in the low-latitudes (Amazon
R1: R=0.90; Amazon R2: R=0.94; Amazon R4: R=0.87; Amazon R5: R=0.77; Congo R6: R=0.91; Congo
R7: R=0.97; Asia R8: R=0.86; Asia R9: R=0.84; **Fig. S5**) compared to higher-latitude areas (Amazon R3:
R=0.60; Amazon R10: R=0.50; **Fig. S5**). This latitudinal gradient was similarly reflected in RMSE values,
with lower errors in equatorial regions (Amazon R1: RMSE=1.78; Amazon R2: RMSE=2.17; Amazon R4:

RMSE=4.67; Congo R6: RMSE=3. 26; Congo R7: RMSE=4.73; Asia R8: RMSE=3.38; Asia R9: RMSE=5.86; **Fig. S6**) versus higher-latitude zones (Amazon R5: RMSE=14.85; Amazon R10: RMSE=6.92; **Fig. S6**).

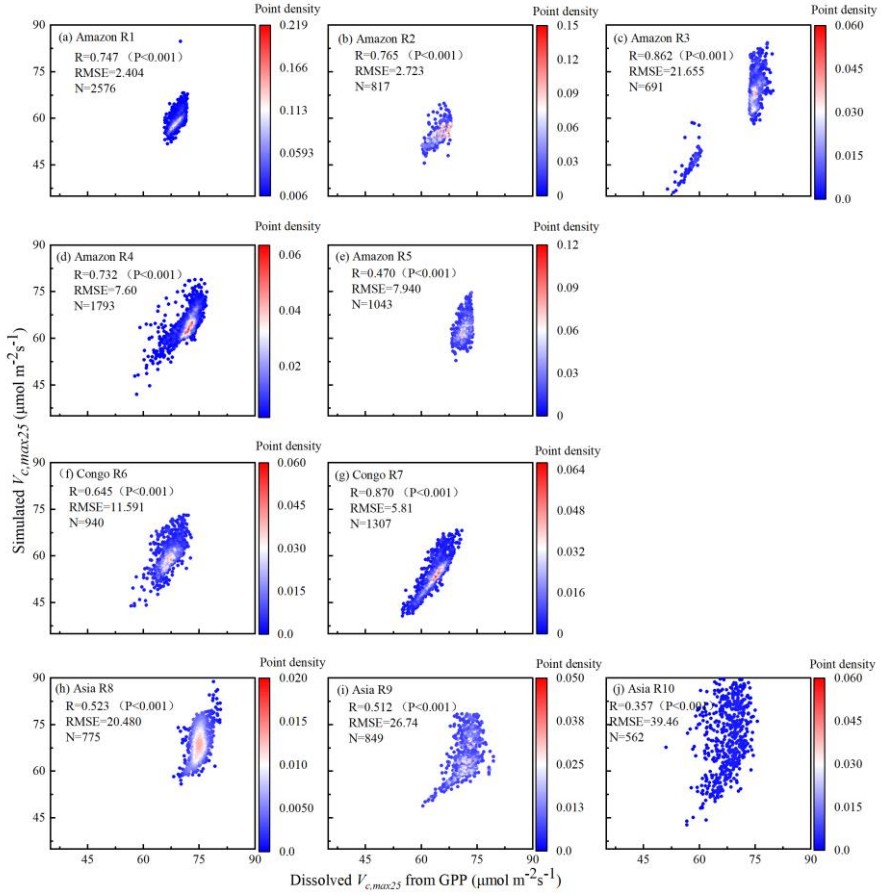

**Figure 4.** Comparisons of the $V_{c,max25}$ of young leaves simulated from RTSIF-derived GPP against that dissolved from GOSIF-derived GPP.

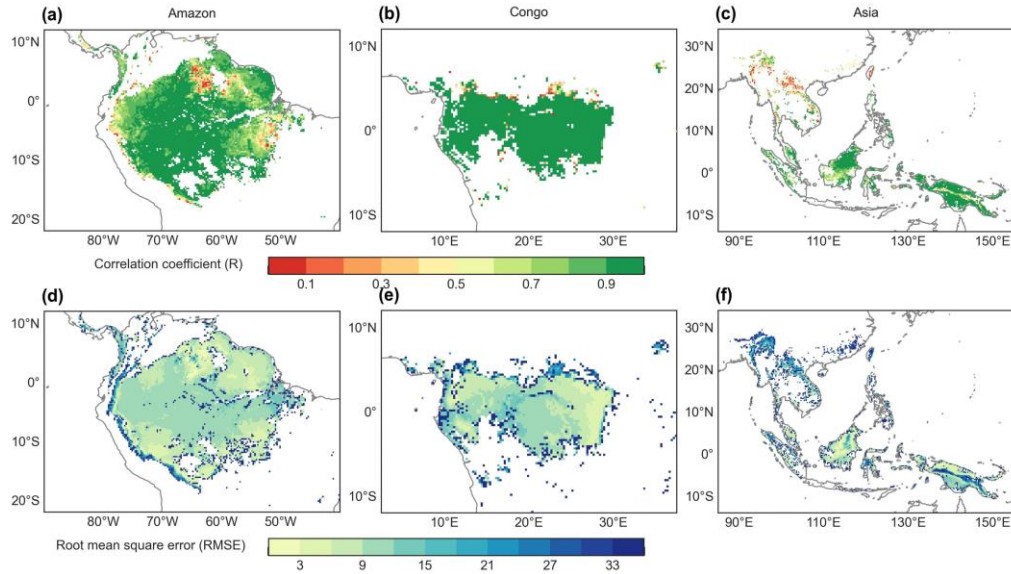

**Figure 5.** The root mean square error (RMSE) and correlation coefficient (R) between the $V_{c,max25}$ of young leaves derived from RTSIF-derived GPP and that dissolved from GOSIF-derived GPP.

### 3.3 Comparison of the seasonal $V_{c,max25}$ of young leaves with the leaf age product

While field measurements have identified distinct seasonal patterns in the $V_{c,max25}$ of young leaves across TEFs, the sparse distribution of observation sites hinders comprehensive assessment of these variations. To address this issue, we conducted K-means clustering analysis on simulated $V_{c,max25}$ maps to evaluate their spatial coherence relative to the leaf age product developed by Yang et al. (2023).

The spatial distribution of clustered $V_{c,max25}$ of young leaves, derived from satellite vegetation signals (**Fig. 6a-c**), closely aligned with climate-based classifications from Chen et al. (2021) (**Fig. 6d-f**). These patterns showed strong correspondence with the Lad-LAI clusters based on endogenous climate variables reported by Yang et al. (2023) (**Fig. 6g-i**). Collectively, these results demonstrate similar spatial clustering patterns. In the middle (R2) and northern (R3) Amazon (**Fig. 7a**), the seasonal variation in the $V_{c,max25}$ of young leaves (**Fig. 8b, c**) was consistent with that of the BR-Sa1 and GF-Guy sites, where young leaves increase during the dry seasons. Moreover, the seasonality of the $V_{c,max25}$ of young leaves in subtropical Asia (**Fig. 8f**) mirrored patterns observed at the CN-Din site, where young leaves conversely increase during the wet seasons. The $V_{c,max25}$ of young leaves peaked in July in sub-region R10, which was located between sub-regions R8 and R9, where the $V_{c,max25}$ of young leaf exhibited a bidirectional phenology (**Fig. 8j**). The four equatorial sub-regions (R1, R2, R7, and R8) exhibited distinct phenological patterns compared to areas near Tropic of Capricorn and Cancer. These equatorial regions demonstrated significantly dampened seasonal variation in the $V_{c,max25}$ of young leaves, with characteristic bidirectional peaks occurring in March and August (**Fig. 8a,d,e,g**). This bimodal pattern contrasts sharply with the unimodal seasonality observed at regions far from equator, consistent with previous findings by Li et al. (2021a).

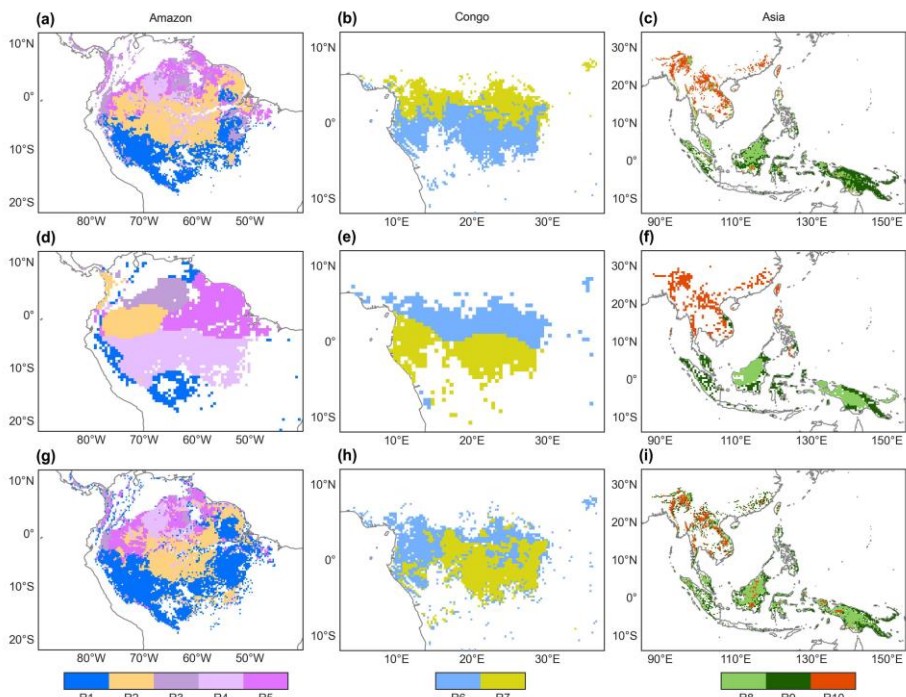

**Figure 6.** Comparison of sub-regions of the $V_{c,max25}$ of young leaves (**a-c**) with those of climatic factors classified by the K-means clustering analysis (**d-f**) analyzed by Chen et al. (2021), and those of the Leaf-age-dependent leaf area index product (Lad-LAI) (**g-i**) developed by Yang et al. (2023).

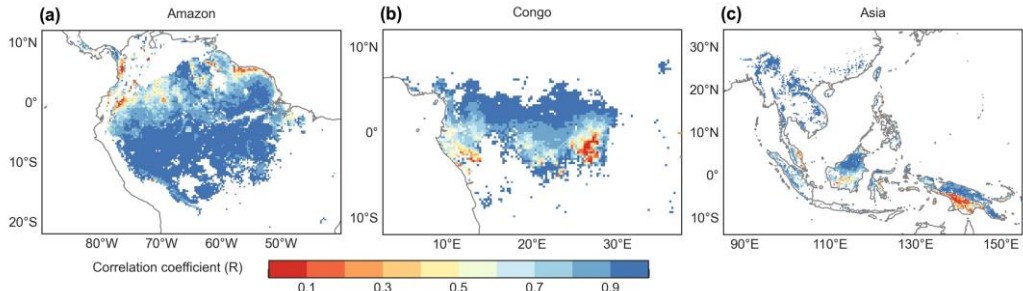

Figure 7. Spatial maps of the correlation coefficient (R) between the monthly simulated $V_{c,max25}$ and the Leaf-age-dependent leaf area index seasonality product (Lad-LAI) developed by Yang et al. (2023).

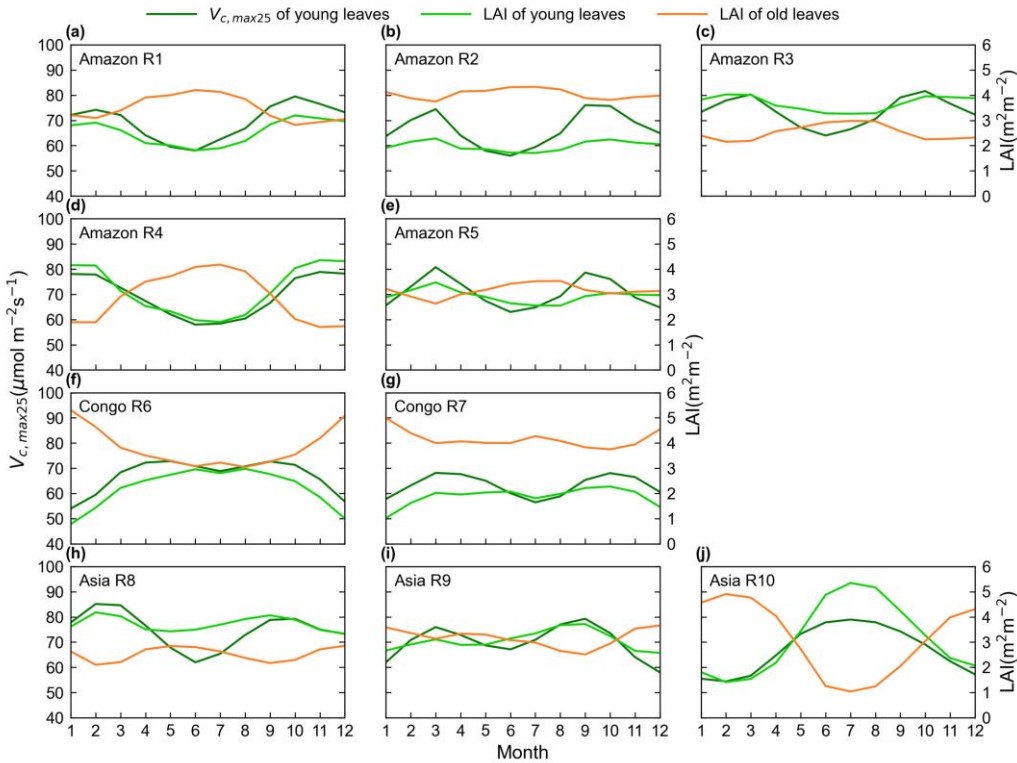

**Figure 8.** Seasonality of the simulated $V_{c,max25}$ of young leaves in comparison with the Leaf-age-dependent leaf area index product (Lad-LAI) developed by Yang et al. (2023).

## 3.4 Partial correlations between the seasonal $V_{c,max25}$ of young leaves and individual climatic factors

To assess the climatic controls on $V_{c,max25}$ of young leaves, we performed spatial partial correlation analyses on climate drivers such as vapor pressure deficit (VPD), air temperature ($T_{mean}$), and downward shortwave solar radiation (SW) (**Fig. 9**), previously established as critical determinants of leaf phenology in TEFs (Yang et al., 2023; Yang et al., 2021; Li et al., 2021a). The $V_{c,max25}$ of young leaves exhibited a strong correlation with the three climate drivers (**Fig. 9**). We then analyzed the relative importance of three climate drivers in influencing $V_{c,max25}$ using the machine-learning model of the Random Forests (RF) method (**Fig. 10, section 2.5.3**). Shortwave radiation exhibited particularly notable positive correlations (R>0.34) with $V_{c,max25}$ across almost all regions with the exception of Amazon sub-region R4 (R=0.17) (**Fig. S8**), and the shortwave radiation was also the most contributing factor (**Fig. 10a**). This underscoring the dominant role of radiation in regulating canopy photosynthesis in TEFs. Although seasonal temperature fluctuations were

modest (**Fig. S7**), likely due to minor temperature gradients, $T_{mean}$ still exhibited a positive correlation with
$V_{c,max25}$ of young leaves. However, at the global scale, $T_{mean}$ had the least influence compared to VPD and
SW (**Fig. 10a**). Notably, in the Asia region, $T_{mean}$ emerged as the primary driver of $V_{c,max25}$ variability and
showed a strong positive correlation in the Asia sub-region R10 (R = 0.88, **Fig. S8**). Notably, VPD and $T_{mean}$
exhibited negative correlations with $V_{c,max25}$ across Congo, with VPD showing a strong negative relationship
in sub-region Congo R7 (R=-0.70) and $T_{mean}$ in sub-region Congo R6 (R=-0.64) (**Fig. S8**). These two factors
primarily governed the spatial variability of $V_{c,max25}$ across the Congo (**Fig. 10 c**). This variability primarily
stems from the canopy turnover patterns, where leaf aging during rainy seasons reverses during dry periods
(Li et al., 2021a; Yang et al., 2023; Yang et al., 2021). As a result, the seasonality of leaf photosynthetic
capacity tended to show an inverse trend to the seasonality of the leaf age, as expected Chen et al. (2020).

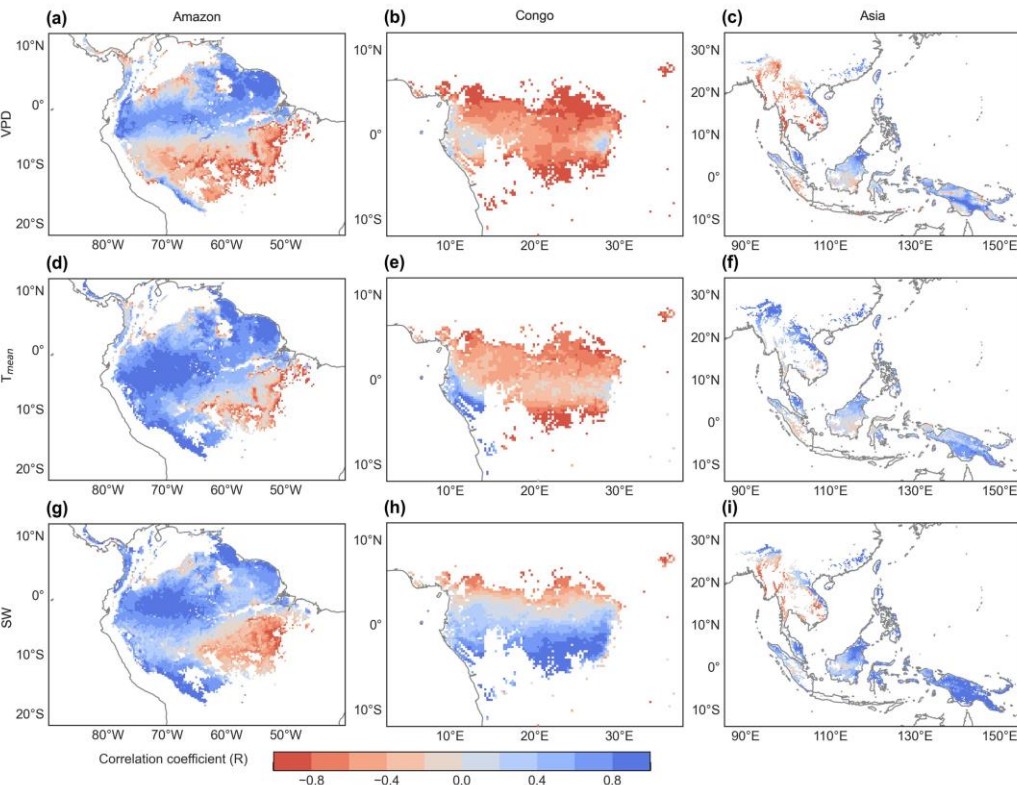


**Figure 9.** Spatial maps of correlation coefficient (R) between the SIF-simulated monthly $V_{c,max25}$ and vapor
pressure deficit (VPD, **a-c**), air temperature ($T_{mean}$, **d-f**), and downward shortwave solar radiation (SW, **g-i**).

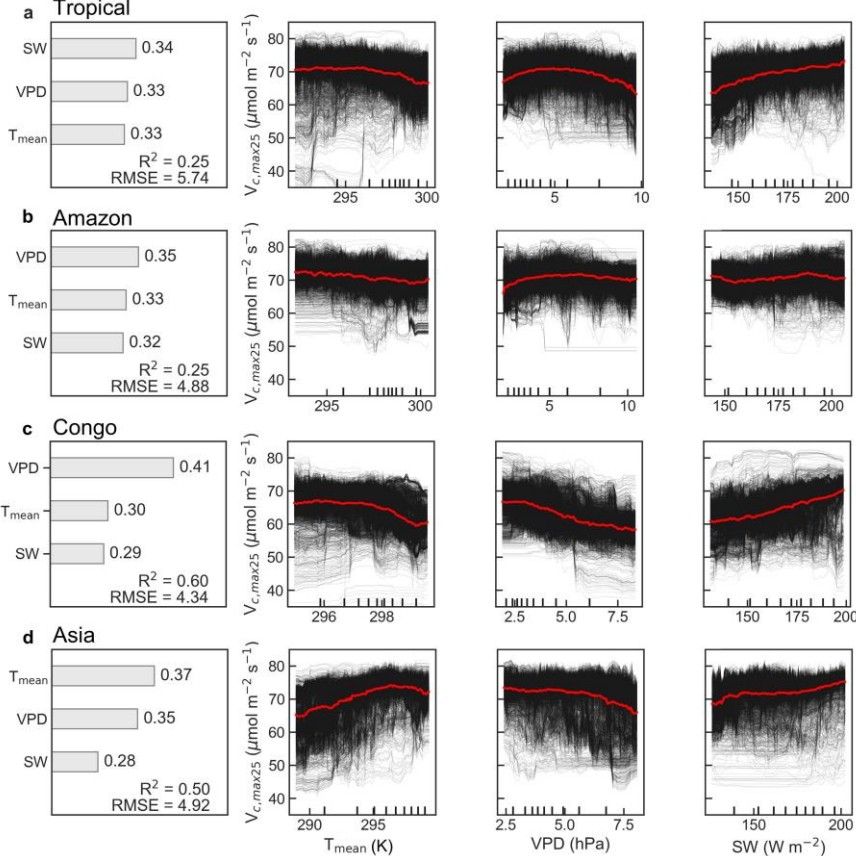


**Figure 10.** Climatic drivers of spatial variations in average $V_{c,max25}$ of young leaves across the TEFs (**a**) and three major tropical forests regions (**b-d**). Contributions (∅) of three climate factors to the multiple-year-average $V_{c,max25}$ using the random forest (RF) algorithm. $R^2$ represents the coefficient of determination between simulated- and observed- $V_{c,max25}$. RMSE indicates the root mean standard error. Partial dependence plots (PDP) of the relationships between three climate drivers [$T_{mean}$ (K), SW (W m$^{-2}$), VPD (hPa)] and $V_{c,max25}$. Relations for each pixel are displayed in black lines and relations on regional average are shown in red lines.

## 3.5 Evaluating potential uncertainties in the $V_{c,max25}$ of young leaves

The seasonal variations in the $V_{c,max25}$ of young leaves using 4×4 neighboring pixels were closely aligned with those observed in the 0.25° products utilizing a grid of 2×2 pixels (**Fig. S9**). Results showed a highly linear correlation between the simulated 0.25° resolution and 0.5°resolution consistent (R>0.99), with the root mean square error (RMSE) being maintained below 0.66 (**Fig. 11**). This evidence demonstrating that the neighbor-based decomposition approach reliably generates the consistent $V_{c,max25}$ products across different spatial scales.

Three distinct versions for the gridded $V_{c,max25}$ of young leaves products from RTSIF- and GOSIF-derived GPP and FLUXCOM GPP at various spatial resolutions (**Figs. S10-S12**) were produced in this study. While minor differences existed among these products, they showed strong spatial consistency (**Fig. 12**) and high similarity in geographic distribution patterns (R: 0.87~0.96, *P*<0.001; **Fig. 13**). All three GPP-derived $V_{c,max25}$ products exhibited consistent seasonal patterns across the ten sub-regions (**Fig. 12**). Validation against *in situ* measurements demonstrated that the RTSIF-derived product achieved optimal performance, showing both the highest correlation (R=0.85) and minimal deviation (RMSE=13.69) with ground observations (**Fig. 13**). These results collectively indicate that the $V_{c,max25}$ of young leaves products reliably capture photosynthetic seasonality across the ten sub-regions.

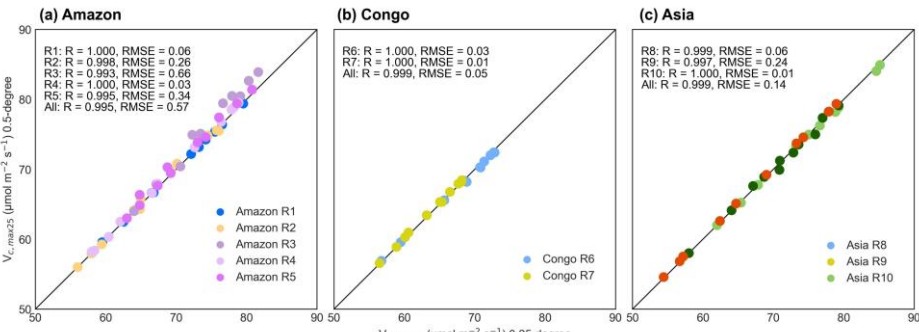


**Figure 11.** Scatter plots between the simulated $V_{c,max25}$ of young leaves simulated using the 2×2 (0.25° resolution) and 4×4 (0.5° resolution) neighboring pixels in the above-mentioned ten clustered sub-regions.

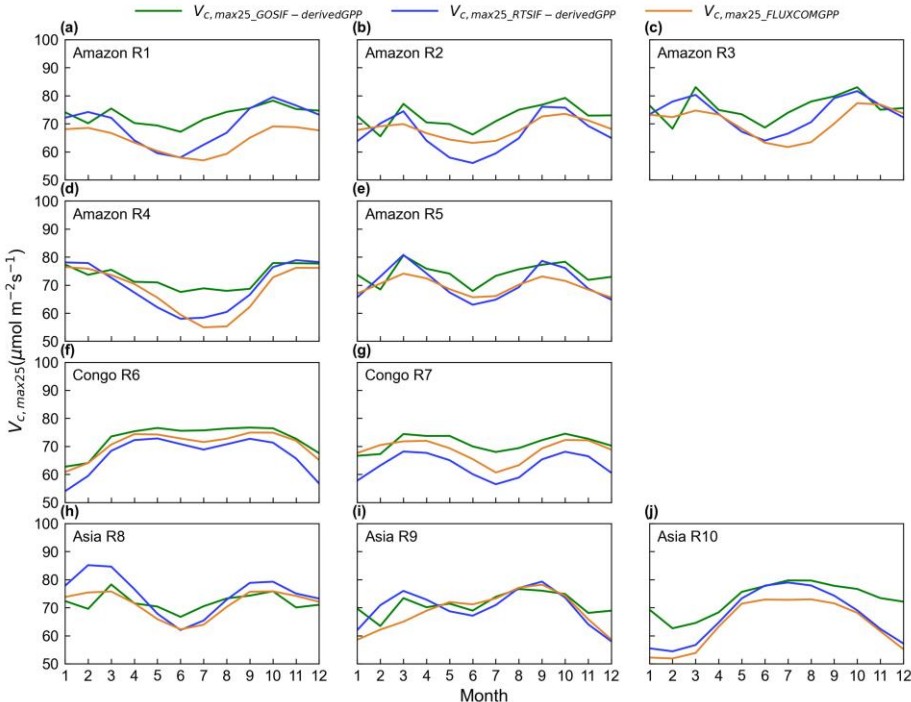


**Figure 12.** Seasonality of the simulated $V_{c,max25}$ of young leaf derived from RTSIF-, GOSIF- and FLUXCOM GPP in the ten clustered sub-regions.


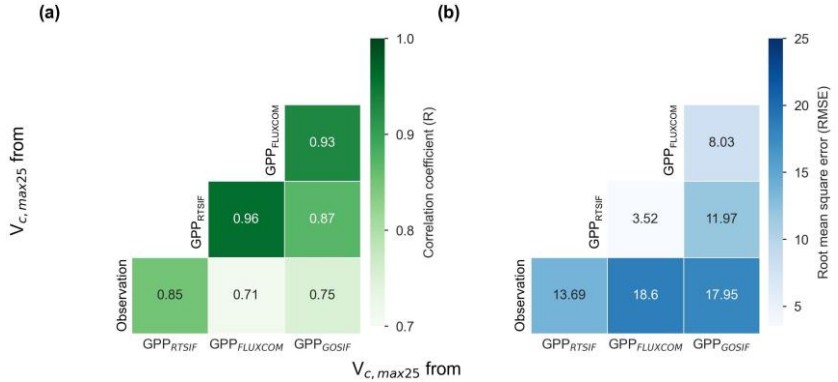


**Figure 13.** Comparison of $V_{c,max25}$ derived from three GPP products (i.e., GPP$_{RTSIF}$, GPP$_{GOSIF}$, GPP$_{FLUXCOM}$) and *in situ* observations. (**a**) Correlation coefficients (R); (**b**) Root mean square error (RMSE).

## 4. Discussion

Tropical forests, marked by no obvious seasonal shifts in greenness but distant variations in leaf age cohorts (Wu et al., 2016; Chen et al., 2020; Chavana-Bryant et al., 2017), show distinct leaf phenology compared with temperate and boreal forests. The young leaves are the main leaf cohort to influence photosynthesis (Oliveira et al., 2023; Sharma et al., 2017; Menezes et al., 2022), as photosynthetic capacity declines with leaf aging (Menezes et al., 2022; Wang et al., 2020). Understanding the leaf age-photosynthesis relationships is therefore critical for assessing plant growth, ecosystem productivity, and carbon cycling in evergreen forests (Albert et al., 2018). The leaf maximum carboxylation capacity ($V_{c,max25}$) serves as a key parameter for modeling photosynthetic $CO_2$ absorption in Earth System Models (ESMs). However, most ESMs typically employ static or annual mean $V_{c,max25}$ values for each plant functional type (Stocker et al., 2020; Atkin et al., 2015). This empirical practice causes uncertainties in tropical forest biomes, which are characterized by their extensive plant functional diversity (Echeverría-Londoño et al., 2018; Spicer et al., 2022) and variable photosynthetic capacity (Piao et al., 2019; Wu et al., 2017a). Furthermore, the $V_{c,max25}$ varies substantially within species due to leaf age, ambient growth temperatures, and the availability of water and nutrients (Stefanski et al., 2020; Lu et al., 2022; Crous et al., 2022). Photosynthesis seasonality in tropical evergreen forests is thus impacted by the replacement of old leaves with younger and more photosynthetically active foliage (Wu et al., 2016; Chen et al., 2020; Chavana-Bryant et al., 2017). These findings underscore the importance of accurately quantifying the leaf age and integrating the leaf age information into $V_{c,max25}$ estimation to enhance simulation of the leaf $CO_2$ assimilation in tropical forests. Currently, there is no comprehensive continental-scale data are available on the leaf age-dependent $V_{c,max25}$ variations throughout tropical evergreen forests. This data gap remains because the insufficient field observations for adequate mapping (Hakala et al., 2019) and limitations in Earth System Models (ESMs) that rely on uncertain climatic parameters (Brunner et al., 2021). These challenges hinder the application of remote sensing and land surface models (LSMs) to accurately model the seasonality of large-scale photosynthesis (Krause et al., 2022).

This study presents a novel continental-scale monthly gridded $V_{c,max25}$ of young leaves. The newly developed dataset was validated at four sites (CN-Din in southern China, MDJ-03 in Congo, BR-Sa1, and GF-Guy in South America) using the field measurements of the $V_{c,max25}$. We assessed the reliability of the gridded $V_{c,max25}$ of young leaf across the entire TEFs through pixel-by-pixel validation against GPP-derived estimates using the dissolved method and leaf age data from Yang et al. (2023). The results reveal substantial age-dependent variation in $V_{c,max25}$ (40-90 μmol m$^{-2}$ s$^{-1}$), consistent with the ranges reported for tropical and subtropical regions in current Earth System Models (Rogers, 2014). These findings highlight the necessity of incorporating leaf-age information in future ESM designs. Moreover, the $V_{c,max25}$ estimates successfully captured the dry-season canopy greening patterns in the north of the equator, demonstrating prominent advances in our ability to promptly monitor the photosynthetic capacity in tropical forests. Both direct and indirect evaluations confirm the robustness of these new photosynthetic products. In equatorial regions with high annual rainfall and minimal dry seasons, canopy phenology exhibits subtler variations compared to those forests near Tropic of Capricorn and Cancer (Yang et al., 2021). The new photosynthetic product successfully captures the characteristic bimodal patterns of $V_{c,max25}$ with limited seasonal amplitude in these areas. To converts the SIF data into GPP, a constant coefficient was used, and $V_{c,max25}$ was assumed to be uniformly distributed across all tropical evergreen forests, potentially introducing further uncertainties. This assumption was reflected in the MSD assessment, where the bias component was predominant, especially near the equator. Nevertheless, the impact of this on the seasonality of photosynthesis was minima; because the phase-dependent component of the RMSE remained relatively insignificant.

The "leaf demographic-identical (LDO)" hypothesis categorizes the leaf cohorts into three distinct age

classes: new leaves (from 1 to 60 days), mature leaves (from 60 to 180 days), and old leaves (larger than 180 days), with corresponding mean $V_{c,max25}$ values as reported by Wu et al. (2016). To enhance comparability between observations and models, we further grouped leaves into two age classes. Leaf ages greater than 6 months are classified as a distinct old leaf class, while leaf ages less than 6 months are combined into a single young leaf class. Menezes et al. (2022) reported that mature leaves (60-180 days) exhibited the highest average $V_{c,max25}$, whereas older leaves (234–612 days) showed lower values (30.4 ± 1.2). The young leaves displayed a 23% higher $V_{c,max25}$ than old leaves, with minimal variation in the latter. Notably, the link between the older leaves and $V_{c,max25}$ remains poorly understood in TEFs due to limited field data (Chen et al., 2020). To address these simulation challenges, we treated $V_{c,max25}$ of old leaves as a static value; potentially introducing errors in photosynthetic performance predictions. This simplification may also affect the accuracy of $V_{c,max25}$ and GPP seasonality in ESMs (De Weirdt et al., 2012). Moreover, additional uncertainties stem from assumptions that neglect the spatial and temporal variations driven by the plant functional type diversity, which shifts with seasonal climate anomalies and high heterogeneity in diverse forest ecosystems. These generalizations could also introduce inaccuracies in simulating seasonal variations in $V_{c,max25}$. Reflecting the inherent variability in photosynthetic behavior across leaf ages, the data revealed two distinct responses: (1) certain species, such as P. tomentosa and P. caimito, exhibited marked reductions in $V_{c,max25}$ with age, whereas (2) others, such as M. angularis and V. parviflora, maintained consistent $V_{c,max25}$ values after reaching their peak. Menezes et al. (2022) identified a modest but significant correlation between the $V_{c,max25}$ and leaf age due to these divergent patterns. Variations in the photosynthetic capacity at the ecosystem level could be influenced by species composition and the distribution of plant functional groups within forests. Furthermore, the seasonal fluctuations in $V_{c,max25}$ of young leaves are closely associated with both plant growth strategies and environmental factors. Higher $V_{c,max25}$ values in young leaves during the early growing season may reflect an adaptive strategy to quickly establish photosynthetic capacity, especially beneficial in competitive environments like tropical and subtropical forests. These seasonal variations directly impact a plant's carbon uptake capacity, potentially leading to increased carbon sequestration within plant biomass and influencing atmospheric $CO_2$ concentrations, which could create feedback loops within the climate system.

In summary, we present a novel approach to develop a gridded dataset that incorporates leaf-age sensitivity into the photosynthesis parameters for TEFs at a continental scale. While some uncertainties persist, we provide a monthly gridded $V_{c,max25}$ of young leaves dataset. This innovation facilitates the comprehensive phenological modeling in ESMs, which typically operate at coarser resolutions. These improvements substantially enhance our ability to monitor and mechanistically interpret the spatiotemporal variations in the $V_{c,max25}$ of young leaves, providing essential data for the parameterization and assessment in ESMs. Furthermore, as remote sensing technologies advance, we anticipate that the enhanced temporal and spatial resolution of RTSIF-derived GPP will facilitate more accurate mapping of the photosynthesis products in future studies.

## 5. Data availability

The 0.25 degree time-series $V_{c,max25}$ data from 2001-2018 is presented in this paper as the main dataset. We also provided another two versions of $V_{c,max25}$ generated from GOSIF-derived GPP and FLUXCOM GPP, respectively. The three datasets are available at https://doi.org/10.5281/zenodo.14807414 (Yang et al., 2025). These datasets are compressed in a GeoTiff format, with a spatial reference of WGS84. Each file in these datasets is named as follows: "V$_{cmax25}$_{GPP source}derived_{YYYYMM}.tif".

## 6. Conclusions

This study develop a novel monthly gridded dataset of $V_{c,max25}$ in combination with ontogeny-dependent

leaf age changes. The new $V_{c,max25}$ of young leaves performs reasonably well in validating against three
independent datasets: including (1) *in situ* observations of the monthly $V_{c,max25}$ records; (2) the $V_{c,max25}$ product
dissolved from the GOSIF-derived GPP; (3) the leaf-age-dependent leaf area index product. Our results
demonstrate that the seasonal dynamics in $V_{c,max25}$ of young leaves are governed by distinct climate-
phenology regimes across tropical and subtropical evergreen broadleaved forests. Specifically, in the central
and southern Amazon, the $V_{c,max25}$ of young leaves decreased during dry season onset (approximately
February) but increased during wet season onset (approximately June). Conversely, the $V_{c,max25}$ of young
leaves in subtropical Asia exhibited peak during the wet season (June or July), coinciding with maximum
rainfall. Near the equator, the $V_{c,max25}$ of young leaves showed a bimodal seasonality with very slight
variations. The $V_{c,max25}$ of young leaves products offer valuable insights into the adaptations of tropical and
subtropical forest to the ongoing climate change, while also serving to improve phenology parameterization
in land surface models (LSMs).
**Supplement.** The supplement related to this article will be available online at once accepted.
**Author contributions.** XC designed the research and wrote the paper. XY and QS wrote the draft,
debugged algorithms and processed data. LH debugged algorithms and reviewed the paper. All the authors
edited and revised the paper.
**Competing interests.** The authors declare no competing interests.
**Acknowledgement.** We would like to thank the editor and reviewers for their valuable time in reviewing
the manuscript.

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
