# Peer review of "Remote sensing of young leaf photosynthetic capacity in tropical and subtropical"

_Earth System Science Data, 2025_

## Referee Comment (RC2)

Maximum carboxylation rate at 25°C (Vc,max25) is a key parameter determines the carbon sequestration rate through photosynthesis. It changes with leaf age and environmental conditions. On the basis of remote sensing data, this study produced the dataset of Vc,max25 in tropical and subtropical evergreen broadleaved forests. This manuscript is interesting and well-written. After some modifications, it is publishable.

**Main concerns**

Sections 3.1-3.3: See all the validation/comparison maps, the dissolved Vc,max25 of young leaves performed consistent in tropical Africa and Asia, but differed a bit more across Amazon region. Please explain.

Section 3.4: Regarding the potential climatic drivers of the seasonality of Vc,max25. The authors only compared their seasonal patterns, while they did not establish an effective statistical model to quantify relationship. I would suggest authors adding such analyses.

Finally, authors should clarified the potential limitations and caveat of the data and method used for mapping the Vc,max25 of young leaves in tropical forests. For instance, please add metric for quality control. And, assuming little seasonal variations of Vc,max25 of old leaves may lead to overestimation/underestimation of the seasonal Vc,max25 of young leaves. In addition, the lack of intensive validations across the pantropical forests may be another limitation.

Other minor comments:

- 1. The manuscript needs substantial review of the English style as there are some language mistakes, which makes the comprehension of the text difficult.
- 2. Line 21-22, Abstract: The research gap is not only the lack of quantification but also the absence of continuous, gridded data covering a large spatial range.
- 3. Line 23, Abstract: "neighborhood pixel" may be as "neighborhood pixels"
- 4. Line 28, Abstract: format R values to two decimal places
- 5. Line 58-60: Perhaps subordinate clauses can be used
- 6. Line 100: Leaves are classified as young or mature based on 180 days, but it needs to be clarified which category includes the 180th day.
- 7. Line 176: The format of Equation (1) may be better like:  $GPP_{total} = LAI_Y \times A_{sat_Y} + LAI_0 \times A_{sat_Q}$
- 8. Line 182: Figure 2. Remove the background color
- 9. Line 201-203: What constraints including in the nonlinear least squares approach?
- 10. Line 381: There is a typo "yong" in the figure
- 11. Lines 163/282/388/409: Change "...the young leaves Vc,max25..."to ".... the Vc,max25 of young leaves..." Please also check other similar mistakes in the manuscript thoroughly.
- 12. Figure 10: The scale of  $T_{air}$  should be appropriately reduced to display seasonal dynamics more effectively.

- 13. Figure S5, S6: Map of Congo in Jun. in Figure S5 should be smaller and this in Oct. in Figure S6 may be not show complete. Please check all maps in supplementary material.
- 14. Line 325-326: 'Keep iterating until there is no change to the centroids. i.e. assignment of data points to clusters isn't changing'. May rephrase.
- 15. Line 342: the mean values (V and U)
- 16. Line 352: the blank between  $5.984^{\circ}$  and S
- 17. Greater attention should be devoted to the details of the figures in the manuscript. For example, The bolded font in Fig1 and Fig 3b. The labels of latitude and longitude in Figures 5-9 should be unified. Please standardize the style of all figures throughout the manuscript, particularly ensuring consistency in the map display, including the axes and other elements.

---

## Author Comment (AC1)

**Responses to reviewer 1#'s comments point by point**

MS No.: essd-2025-64 Title: Remote sensing of young leaf photosynthetic capacity in tropical and subtropical evergreen broadleaved forests Author(s): Xueqin Yang et al.

**General Comments of Reviewer 1#:**

The manuscript presents a significant advancement in understanding the photosynthetic capacity of young leaves in tropical and subtropical evergreen broadleaved forests through a novel satellite-based approach to estimate Vc,max25. The proposed approach for deriving Vc,max25 is well-constructed and contributes to filling a critical gap in our understanding of leaf age and its impact on photosynthetic efficiency. A few minor revisions could improve the clarity and completeness of the manuscript:

**Response:** We appreciate the time and efforts of the editor and referees in reviewing this manuscript and the valuable suggestions offered. Please see our response to your comments in the supplement below.

**Minor Comments:**

**Comment 1:** 1. While the approach for deriving Vc,max25 from SIF data is compelling, the assumption of a constant Vc,max25 for old leaves could benefit from further explanation.

**Response:** Thank you for the positive comments on the novelty of our proposed dataset. We agree with the reviewer that it is a necessary to provide more explanation about the assumption of a constant  $V_{c,max25}$  for old leaves. We have referenced additional literature. Existing studies (Niinemets et al., 2015; Kitajima et al., 1997; Yoder et al., 1994) suggest that the photosynthetic capacity of old leaves in tropical evergreen forests tends to remain relatively stable over time, especially compared to young leaves, which exhibit more pronounced seasonal fluctuations. While this assumption may introduce some bias, it enables us to focus on the dominant seasonal patterns driven by young leaves, which have been shown to play a key role in the overall canopy photosynthesis. In this study, we set a constant value of  $V_{c,max25} = 20 \ \mu mol \ m^{-2} \ s^{-1}$  for old leaves, derived from the asymptotic trend between leaf ag and  $V_{c,max}$  (**Figure R1**). This value reflects the stabilization of photosynthetic capacity at a low level once leaves reach their old-age stage.

*Figure R1* Vc,max (**a**) and Relative leaf efficiency (erel) as a function of relative leaf age (arel) (**b**) (cf. Chen et al., 2020).

**Reference:**

- Chen, X., Maignan, F., Zhang, Y., Viovy, N., Bastos, A., Liu, L., Goll, D., Wu, J., Liu, L.
  Y., Yue, C., Peng, S. S., Yuan, W. P., da Conceicao, A. C., O'Sullivan, M., and Ciais,
  P.: Novel Representation of Leaf Phenology Improves Simulation of Amazonian Evergreen Forest Photosynthesis in a Land Surface Model, J. Adv. Model. Earth Syst., 12, e2018MS001565, https://doi.org/10.1029/2018ms001565, 2020.
- Kitajima, K., Mulkey, S. S., and Wright, S. J.: Decline of photosynthetic capacity with leaf age in relation to leaf longevities for five tropical canopy tree species. American Journal of Botany, 84(5), 702-708. 1997.
- Niinemets, Ü., Cescatti, A., Rodeghiero, M. and Tosens, T.: Leaf internal diffusion conductance limits photosynthesis more strongly in older leaves of Mediterranean evergreen broad-leaved species. Plant, Cell & Environ., 28: 1552-1566. https://doi.org/10.1111/j.1365-3040.2005.01392.x. 2015
- Yoder, B. J., Ryan, M. G., Waring, R. H., Schoettle, A. W., and Kaufmann, M. R.: Evidence of reduced photosynthetic rates in old trees. Forest Science, 40(3), 513-527. 1994

**Comment 2:** 2. The results show interesting seasonal trends in young leaf Vc,max25. It would be useful to discuss the ecological implications of these seasonal variations in the context of the carbon cycle.

**Response:** Thank you very much for your insightful comments on the seasonal trends in  $V_{c,max25}$  of young leaves. We agree that discussing the ecological implications of these seasonal variations in the context of the carbon cycle would be highly valuable.

The seasonal trends in  $V_{cmax,25}$  of young leaves is indeed intriguing and likely related to both plant growth strategies and environmental factors. The relatively higher  $V_{c,max25}$  in young leaves during the early growing season may be an adaptive strategy for plants to rapidly establish their photosynthetic capacity, allowing them to make the most of favorable light and temperature conditions and giving them a competitive edge in highly competitive environments such as tropical and subtropical forests. These variations may also be closely linked to seasonal fluctuations in environmental factors, with plants adjusting their  $V_{c,max25}$  to cope with water stress during the dry season and maximizing photosynthetic efficiency during the rainy season. At the ecosystem level, these seasonal variations in young leaf  $V_{c,max25}$  directly influence a plant's carbon uptake capacity, potentially leading to more carbon being fixed within plant biomass and affecting atmospheric CO2 concentrations. This, in turn, could create feedback loops within the climate system and interact with other ecological processes such as soil carbon cycling and microbial activities.

We added some expand in section 4 as follows:

"Furthermore, the seasonal fluctuations in  $V_{c,max25}$  of young leaves are closely associated with both plant growth strategies and environmental factors. Higher  $V_{c,max25}$  values in young leaves during the early growing season may reflect an adaptive strategy

to quickly establish photosynthetic capacity, especially beneficial in competitive environments like tropical and subtropical forests. These seasonal variations directly impact a plant's carbon uptake capacity, potentially leading to increased carbon sequestration within plant biomass and influencing atmospheric CO2 concentrations, which could create feedback loops within the climate system" (**In revision lines 580– 586**)

**Comment 3:** 3 The authors could briefly discuss the limitations of the proposed method, particularly in regions with high cloud cover or in areas where SIF data quality might be compromised. This would help users of the dataset understand its potential applications and limitations in various settings.

**Response:** Thank you for your insightful comments. High cloud cover can lead to reduced quality in optical remote sensing data, or poor SIF data quality itself, both of which can impact the accuracy of  $V_{c,max25}$  estimations. We have carefully considered your suggestions and have incorporated a detailed quality control (QC) metric to ensure the reliability of our methodology and prevent potential misuse of the data.

We provided information of data quality control (QC) for the  $V_{c,max25}$  of young leaves product to prevent data misuse. In the QC system (**Table S5**), data quality is divided into four levels: level 1 represents the highest quality; level 2 and level 3 represent good and acceptable quality, respectively; and level 4 warns to be used cautiously. This QC product is generated based on Pearson's correlation coefficients (R) and the root mean square error (RMSE), which were obtained by comparing the seasonal  $V_{c,max25}$  estimated from RTSIF- and GOSIF-derived GPP. Results showed that more than 91.5% of pixels are with QC at best and only less than 0.03% are with QC at level 3 and level 4. These details are elaborated in **section 2.6** (**In revision lines 350-356**).

**2.6** *Quality control (QC) for young leaves Vc,max25 product**

We provided information on data quality control (QC) along with the  $V_{c,max25}$  of young leaves product. In the QC system (**Table S5**), data quality was divided into four levels: Level 1 represents the highest quality, Level 2 and Level 3 represent good and acceptable quality, respectively, and Level 4 should be used with caution. This QC product was generated based on Pearson's correlation coefficients (R) and the root mean square error (RMSE), which were obtained by comparing the seasonal  $V_{c,max25}$ estimated from RTSIF- and GOSIF-derived GPP.

| e so momation of data quanty control (QC) for the re,max25 product |                   |                |                      |                                              |
|--------------------------------------------------------------------|-------------------|----------------|----------------------|----------------------------------------------|
|                                                                    | QC class          | QC value       | R                    | RMSE (µmol m -2 s -1 ) |
|                                                                    | Best              | 1              | <mark>0.6-1</mark>   | <mark>0-10</mark>                            |
|                                                                    | <mark>Good</mark> | <mark>2</mark> | <mark>0.4-0.6</mark> | <mark>10-20</mark>                           |
|                                                                    | Acceptable        | <mark>3</mark> | <mark>0.2-0.4</mark> | <mark>20-30</mark>                           |
|                                                                    | Cautious use      | <mark>4</mark> | <mark><0.2</mark> | <mark>>30</mark>                          |

**Table S5** Information of data quality control (OC) for the Vc.max25 product

---

## Author Comment (AC2)

**Responses to reviewer 2#'s comments point by point**

MS No.: essd-2025-64
Title: Remote sensing of young leaf photosynthetic capacity in tropical and subtropical evergreen broadleaved forests
Author(s): Xueqin Yang et al.

**General Comments of Reviewer 2#:**

Maximum carboxylation rate at 25°C (Vc,max25) is a key parameter determines the carbon sequestration rate through photosynthesis. It changes with leaf age and environmental conditions. On the basis of remote sensing data, this study produced the dataset of Vc,max25 in tropical and subtropical evergreen broadleaved forests. This manuscript is interesting and well-written. After some modifications, it is publishable.

*Response: We appreciate the time and efforts of the editor and referees in reviewing this manuscript and the valuable suggestions offered. Please see our response to your comments in the supplement below.*

**Main concerns:**

**Comment 1:**

Sections 3.1-3.3: See all the validation/comparison maps, the dissolved Vc,max25 of young leaves performed consistent in tropical Africa and Asia, but differed a bit more across Amazon region. Please explain.

*Response: Thanks for your careful review. **Figure 4** shows that $V_{c,max25}$ in the Amazon region exhibits a highly clustered distribution across sub-regions, whereas in the Congo and Asia regions, it follows a more continuous gradient pattern. To investigate the underlying causes, we analyzed the spatial distribution of plant functional types (PFTs) in tropical regions, focusing on the EBF fraction at the pixel level. As shown in **Figure R2**, more than 69.78% of the Amazon region has an EBF fraction exceeding 90%, compared to only 41.09% in Congo and 31.45% in tropical Asia. This indicating higher pixel purity and reduced canopy variability, which contributes to the more concentrated range of $V_{c,max25}$ values. Furthermore, a comparison with the dry season onset map from Tang and Dubayah (2017) reveals a similar spatial pattern (**Figure R3**), suggesting that the identified sub-regions correspond to relatively homogeneous environmental conditions. This environmental uniformity results in similar plant growth conditions, leading to a more constrained $V_{c,max25}$ distribution. Notably, sub-region A3, located in the northwestern Amazon near coastal and mountainous areas, forms two distinct clustering zones.*

[Figure]

*Figure R2 Spatial pattern of evergreen broadleaf forests (EBF) fraction across tropical*

[Figure]

*Figure R3 Climate zones in Amazon identified according to start of dry season (cf. Tang and Dubayah, 2017)*

*We have added further explanations in **section 3.2** to clarify the differences between the Amazon, and Congo, Asia regions:*

*"Across the Amazon, more than 69.78% of pixels have a high EBF fraction (>90%). The spatial clustering pattern aligns with the onset of the dry season (cf. Tang and Dubayah, 2017), suggesting that the clustering analysis effectively differentiates climate regions within the Amazon. The relatively homogeneous environmental conditions across these sub-regions create similar plant growth environments, leading to a more constrained range of $V_{c,max25}$ values and pronounced clustering effects in sub-regions A1–A5. Notably, sub-region A3, located in the northwestern Amazon near coastal and mountainous areas, forms two distinct clustering zones." (**In revision lines 396-402**)*

**Comment 2:**

Section 3.4: Regarding the potential climatic drivers of the seasonality of $V_{c,max25}$. The authors only compared their seasonal patterns, while they did not establish an effective statistical model to quantify relationship. I would suggest authors adding such analyses.

*Response: Thanks so much for pointing out this important question. We totally agree and thus did additional analyses completely following the reviewer's suggestions.*

*Regarding the potential climatic drivers of $V_{c,max25}$ seasonality, we actually establish a Random Forests (RF) model for entire tropical as well as for three major tropical forest regions to quantify the relationship between $V_{c,max25}$ and three climate variables (SW, VPD and $T_{mean}$). The results are presented in Figure 10 in the revised manuscript. Our findings reveal substantial regional differences in climatic drivers, with VPD primarily regulating $V_{c,max25}$ in the Amazon and Congo regions, while $T_{mean}$ plays a dominant role in Asia. However, at the tropical scale, SW emerges as the key primary controlling factor. To further quantify the relationships, we systematically analyzed the potential climatic drivers of $V_{c,max25}$ seasonality across the 10 sub-regions identified in the clustering analysis. We have revised **section 3.4** as follows:*

**"3.4 Partial correlations between the seasonal $V_{c,max25}$ of young leaves and individual climatic factors**

*To assess the climatic controls on $V_{c,max25}$ of young leaves, we performed spatial partial correlation analyses on climate drivers such as vapor pressure deficit (VPD), air temperature ($T_{mean}$), and downward shortwave solar radiation (SW) (**Fig. 9**), previously established as critical determinants of leaf phenology in TEFs (Yang et al., 2023; Yang et al., 2021; Li et al., 2021a). The $V_{c,max25}$ of young leaves exhibited a strong correlation with the three climate drivers (**Fig. 9**). We then analyzed the relative importance of three climate drivers in influencing $V_{c,max25}$ using the machine-learning model of the Random Forests (RF) method (**Fig. 10, section 2.5.3**). Shortwave radiation exhibited particularly notable positive correlations (R>0.34) with $V_{c,max25}$ across almost all regions with the exception of Amazon sub-region R4 (R=0.17) (**Fig. S8**), and the shortwave radiation was also the most contributing factor (**Fig. 10a**). This underscoring the dominant role of radiation in regulating canopy photosynthesis in TEFs. Although seasonal temperature fluctuations were modest (**Fig. S7**), likely due to minor temperature gradients, $T_{mean}$ still exhibited a positive correlation with $V_{c,max25}$ of young leaves. However, at the global scale, $T_{mean}$ had the least influence compared to VPD and SW (**Fig. 10a**). Notably, in the Asia region, $T_{mean}$ emerged as the primary driver of $V_{c,max25}$ variability and showed a strong positive correlation in the Asia sub-region R10 (R = 0.88, **Fig. S8**). Notably, VPD and $T_{mean}$ exhibited negative correlations with $V_{c,max25}$ across Congo, with VPD showing a strong negative relationship in sub-region Congo R7 (R=-0.70) and $T_{mean}$ in sub-region Congo R6 (R=-0.64) (**Fig. S8**). These two factors primarily governed the spatial variability of $V_{c,max25}$ across the Congo (**Fig. 10 c**). This variability primarily stems from the canopy turnover patterns, where leaf aging during rainy seasons reverses during dry periods (Li et al., 2021a; Yang et al., 2023; Yang et al., 2021). As a result, the seasonality of leaf photosynthetic capacity tended to show an inverse trend to the seasonality of the leaf age, as expected Chen et al. (2020)."* **(In revision lines 454-475)**

[Figure]

**Figure 9.** *Spatial maps of correlation coefficient (R) between the SIF-simulated monthly $V_{c,max25}$ and vapor pressure deficit (VPD, **a-c**), air temperature ($T_{mean}$, **d-f**), and downward shortwave solar radiation (SW, **g-i**).*

[Figure]

**Figure 10.** *Climatic drivers of spatial variations in average $V_{c,max25}$ of young leaves across the TEFs (**a**) and three major tropical forests regions (**b-d**). Contributions (∅) of three*

*climate factors to the multiple-year-average $V_{c,max25}$ using the random forest (RF) algorithm. $R^2$ represents the coefficient of determination between simulated- and observed- $V_{c,max25}$. RMSE indicates the root mean standard error. Partial dependence plots (PDP) of the relationships between three climate drivers [$T_{mean}$ (K), SW (W m$^{-2}$), VPD (hPa)] and $V_{c,max25}$. Relations for each pixel are displayed in black lines and relations on regional average are shown in red lines.*

**Comment 3:**

Finally, authors should clarified the potential limitations and caveat of the data and method used for mapping the Vc,max25 of young leaves in tropical forests. For instance, please add metric for quality control. And, assuming little seasonal variations of Vc,max25 of old leaves may lead to overestimation/underestimation of the seasonal Vc,max25 of young leaves. In addition, the lack of intensive validations across the pantropical forests may be another limitation.

***Response:*** *Thank you for your insightful comments. We have carefully considered your suggestions and have made the following revisions to address the concerns raised: (1) we have incorporated a detailed quality control (QC) metric to ensure the reliability of our methodology and prevent potential misuse of the data. The QC data are provided in **section 2.6**. (2) We have added statements and discussion in **section 4** to acknowledge the potential limitations arising from assuming minimal seasonal variation in $V_{c,max25}$ of old leaves and the lack of extensive validation across pantropical forests.*

- **add quality control (QC) metric to prevent potential misuse**

*We provided information of data quality control (QC) for the $V_{c,max25}$ of young leaves product to prevent data misuse. In the QC system (**Table S5**), data quality is divided into four levels: level 1 represents the highest quality; level 2 and level 3 represent good and acceptable quality, respectively; and level 4 warns to be used cautiously. This QC product is generated based on Pearson's correlation coefficients (R) and the root mean square error (RMSE), which were obtained by comparing the seasonal $V_{c,max25}$ estimated from RTSIF- and GOSIF-derived GPP. Results showed that more than 91.5% of pixels are with QC at best and only less than 0.03% are with QC at level 3 and level 4. These details are elaborated in **section 2.6 (In revision lines 350-356)**.*

**2.6 Quality control (QC) for young leaves $V_{c,max25}$ product**

*We provided information on data quality control (QC) along with the $V_{c,max25}$ of young leaves product. In the QC system (**Table S5**), data quality was divided into four levels: Level 1 represents the highest quality, Level 2 and Level 3 represent good and acceptable quality, respectively, and Level 4 should be used with caution. This QC product was generated based on Pearson's correlation coefficients (R) and the root mean square error (RMSE), which were obtained by comparing the seasonal $V_{c,max25}$ estimated from RTSIF- and GOSIF-derived GPP.*

**Table S5** Information of data quality control (QC) for the $V_{c,max25}$ product

| QC class | QC value | R | RMSE ($\mu$mol m$^{-2}$ s$^{-1}$) |
|---|---|---|---|
| Best | 1 | 0.6-1 | 0-10 |
| Good | 2 | 0.4-0.6 | 10-20 |

| | | | |
|---|---|---|---|
| Acceptable | 3 | 0.2-0.4 | 20-30 |
| Cautious use | 4 | <0.2 | >30 |

- **add caveat statement to warn potential uncertainties**

We have added caveat statements and uncertainties in **section 4** to discuss the potential limitations of our assumptions and the constraints of data validation. In the **section 4**, some revised as follows:

"Notably, the link between the older leaves and $V_{c,max25}$ remains poorly understood in TEFs due to limited field data (Chen et al., 2020). To address these simulation challenges, we treated $V_{c,max25}$ of old leaves as a static value; potentially introducing errors in photosynthetic performance predictions. This simplification may also affect the accuracy of $V_{c,max25}$ and GPP seasonality in ESMs (De Weirdt et al., 2012). Moreover, additional uncertainties stem from assumptions that neglect the spatial and temporal variations driven by the plant functional type diversity, which shifts with seasonal climate anomalies and high heterogeneity in diverse forest ecosystems. These generalizations could also introduce inaccuracies in simulating seasonal variations in $V_{c,max25}$. Reflecting the inherent variability in photosynthetic behavior across leaf ages, the data revealed two distinct responses: (1) certain species, such as P. tomentosa and P. caimito, exhibited marked reductions in $V_{c,max25}$ with age, whereas (2) others, such as M. angularis and V. parviflora, maintained consistent $V_{c,max25}$ values after reaching their peak. Menezes et al. (2022) identified a modest but significant correlation between the $V_{c,max25}$ and leaf age due to these divergent patterns. Variations in the photosynthetic capacity at the ecosystem level could be influenced by species composition and the distribution of plant functional groups within forests." **(In revision lines 566-579)**

"The new photosynthetic product successfully captures the characteristic bimodal patterns of $V_{c,max25}$ with limited seasonal amplitude in these areas. To converts the SIF data into GPP, a constant coefficient was used, and $V_{c,max25}$ was assumed to be uniformly distributed across all tropical evergreen forests, potentially introducing further uncertainties. This assumption was reflected in the MSD assessment, where the bias component was predominant, especially near the equator. Nevertheless, the impact of this on the seasonality of photosynthesis was minima; because the phase-dependent component of the RMSE remained relatively insignificant." **(In revision lines 552-558)**

**Minor Comments:**

**Comment 1:** 1. The manuscript needs substantial review of the English style as there are some language mistakes, which makes the comprehension of the text difficult.

*Response: We agree with the reviewer that it is essential to ensure that our manuscript is written clearly and effectively in English. We conducted a thorough review of our manuscript to address any language mistakes and improve the overall readability of the text.*

**Comment 2:** 2. Line 21-22, Abstract: The research gap is not only the lack of quantification but also the absence of continuous, gridded data covering a large spatial range.

*Response: We completely agree with the reviewer's comment and revised the argument as suggested:*

*"Nevertheless, quantifying the leaf photosynthetic capacity of different age across TEFs remains challenging, especially when considering continuous temporal variations at continental scales" **(In revision lines 22-23)***

**Comment 3:** 3. Line 23, Abstract: "neighborhood pixel" may be as "neighborhood pixels".

*Response: Thanks, we revised it as suggested. **(In revision line 24)***

**Comment 4:** 4. Line 28, Abstract: format R values to two decimal places.

*Response: Many thanks. The revised sentence was as follows:*

*"Validations against in situ observations demonstrate that the newly developed $V_{c,max25}$ products accurately capture the seasonality of young leaves in South America and subtropical Asia, with correlation coefficients of 0.84, 0.66, and 0.95, respectively." **(In revision lines 28-30)***

**Comment 5:** 5. Line 58-60: Perhaps subordinate clauses can be used.

*Response: Thanks for your careful review. The revised sentence was as follows:*

*"Research on this issue remains limited and inconclusive, largely due to the complex interplay of seasonal constraints such as water availability and light, which affect leaf flushing and shedding processes across different climatic zones (Brando et al., 2010; Yang et al., 2021)." **(In revision lines 57-60)***

**Comment 6:** 6. Line 100: Leaves are classified as young or mature based on 180 days, but it needs to be clarified which category includes the 180th day.

*Response: Thanks for your valuable suggestions. The 180th day is classified to young leaves in this study. We modified the description as follows:*

*"To address the aforementioned gaps in mapping $V_{c,max25}$ of young leaves, we categorized the canopy foliage of TEFs into two distinct leaf age groups: young (≤180 days) and old (>180 days) leaves." **(In revision lines 96-97)***

**Comment 7:** 7. Line 176: The format of Equation (1) may be better like: $GPP_{total} = LAI_Y \times A_{sat\_Y} + LAI_O \times A_{sat\_O}$

*Response: Thanks for your carefulness and nice suggestions. We have revised the Equation (1) and the related description as follows:*

*"The total GPP was simulated using the FvCB photochemical model by combining the LAI groups (young leaf $LAI_Y$ vs. old leaf $LAI_O$; **Equation 1**) and the corresponding net assimilation rates of $CO_2$ (young and mature $An_{sat\_Y}$ vs. old leaf $An_{sat\_O}$; **Equation 1**) (Farquhar et al., 1980).*

$$GPP_{total} = LAI_Y \times An_{sat\_Y} + LAI_O \times An_{sat\_O} \qquad (1)$$

*where $LAI_Y$ represents the LAI of young leaves (≤180 days) and $LAI_O$ represents the LAI of old leaves (>180 days). $An_{sat\_Y}$ and $An_{sat\_O}$ represent the net $CO_2$ assimilation rates of young and old leaves, respectively. The sum of $LAI_Y$ and $LAI_O$ was set as the total*

*canopy LAI. GPP_{total} refers to the total gross primary production of the canopy." (**In revision lines 166-173**)*

**Comment 8:** 8. Line 182: Figure 2. Remove the background color

*Response: Many thanks. We removed the background color to clearly display the procedures as follows:*

[Figure]

***Figure 2.*** *Procedures for mapping the $V_{c,max25}$ of young leaves using a neighbor-based approach.*

**Comment 9:** 9. Line 201-203: What constraints including in the nonlinear least squares approach?

*Response: Thank you for raising this important question. In the nonlinear least squares, two constraints were applied: (1) the $V_{c,max25}$ must be positive ($V_{c,max25}>0$), and (2) the optimal $V_{c,max25}$ was determined by minimizing the residual. Further details on these constraints are added in **section 2.4 (Methods for simulating the $V_{c,max25}$ of young leaves)** of the revised manuscript.*

*"The LAI and $V_{c,max25}$ of young leaves were estimated using nonlinear least squares and constraints on the basis of the GPP values with the four neighboring pixels according to **Equation 1**. The optimal $V_{c,max25}$ was determined by minimizing the residual while satisfying the positivity constraint (i.e., $V_{c,max25}>0$)." (**In revision lines 192-195**)*

**Comment 10:** 10. Line 381: There is a typo "yong" in the figure

*Response: Thanks for your carefulness suggestions. We have corrected the legend as "Simulated $V_{c,max25}$ of young leaves".*

[Figure]

**Figure 3.** *Validations of simulated seasonal $V_{c,max25}$ for all canopy leaves and young leaves with in situ observations. The green lines and green dots are the seasonal young leaf $V_{c,max25}$ simulated from RTSIF derived GPP by the proposed method. The black line and red dots are the mean leaf age $V_{c,max25}$ values from the simulations and in situ observations, respectively. Simulated $V_{c,max25}$ denoted as the young leaf $V_{c,max25}$ simulated from RTSIF-derived GPP by using the new proposed method. Mean $V_{c,max25}$ denoted as the mean leaves age $V_{c,max25}$.*

**Comment 11:** 11. Lines 163/282/388/409: Change "…the young leaves Vc,max25…"to "…. the Vc,max25 of young leaves…" Please also check other similar mistakes in the manuscript thoroughly.

*Response: Thank you for these valuable suggestions. We have replaced the description (i.e., the young leaves $V_{c,max25}$) with your suggestion (i.e., the $V_{c,max25}$ of young leaves). To be cautious, we also corrected other sentences with similar grammar issues in revised manuscript.*

*Line 163 (**In revision line 156**): "2.4 Methods for simulating the $V_{c,max25}$ of young leaves"*

*Line 282 (**In revision line 276**): "2.5 Methods for evaluating the simulated the $V_{c,max25}$ of young leaves"*

*Line 388 (**In revision lines 393-394**): "3.2 Validation of the $V_{c,max25}$ of young leaves simulated from RTSIF-derived GPP against that dissolved from GOSIF-derived GPP"*

*Line 409 (**In revision lines 415-416**): "Figure 4 Comparisons of the $V_{c,max25}$ of young leaves simulated from RTSIF-derived GPP against that dissolved from GOSIF-derived*

*GPP"*

**Comment 12:** 12. Figure 10: The scale of T air should be appropriately reduced to display seasonal dynamics more effectively.

*Response: Thanks for your careful review. We revised the scale to clearly display the seasonal dynamics of climatic variables.*

[Figure]

***Figure S7.*** *Seasonality of $V_{c,max25}$ of young leaves, air temperature ($T_{air}$), vapor pressure deficit (VPD) and downward shortwave solar radiation (SW) in the ten sub-regions classified using the K-means clustering analysis method.*

**Comment 13:** 13. Figure S5, S6: Map of Congo in Jun. in Figure S5 should be smaller and this in Oct. in Figure S6 may be not show complete. Please check all maps in supplementary material.

*Response: Thanks for your careful review. In the revised manuscript, we reproduced Figs. S8-10 (in revision **Figs. S10-S12**) with unified format to eliminate the inconsistencies of spatial scale. To be cautious, we also reproduced and carefully checked all maps in revised manuscript.*

[Figure]

*Figure S10*. Spatial distributions and seasonal changes of young leaf $V_{c,max25}$ in TEFs derived from RT-SIF (2001–2018). White areas are missing data.

[Figure]

*Figure S11*. Spatial distributions and seasonal changes of young leaf $V_{c,max25}$ in TEFs derived from Go-SIF (2001–2018). White areas are missing data.

[Figure]

*Figure S12*. *Spatial distributions and seasonal changes of young leaf $V_{c,max25}$ in TEFs derived from FLUXCOM (2001–2013). White areas are missing data. Black dots are invalid value.*

**Comment 14:** 14. Line 325-326: 'Keep iterating until there is no change to the centroids. i.e. assignment of data points to clusters isn't changing'. May rephrase.

*Response: Many thanks. We have implemented comprehensive revisions, with particular attention to lexical precision and grammatical accuracy throughout the document. As suggested, we rephrased this sentence as follows:*

*"Iterate until convergence (i.e., cluster assignments remain unchanged between iterations)."* **(In revision line 317)**

**Comment 15:** 15. Line 342: the mean values (V and U).

*Response: Thanks, we revised it as suggested.*

*"...and $\bar{V}$ and $\bar{U}$ are the mean values of the simulated and observed in situ measurements $V_{c,max25}$."* **(In revision lines 343-344)**

**Comment 16:** 16. Line 352: the blank between 5.984° and S.

*Response: Many thanks. We deleted the blank and thoroughly proofread the manuscript to correct the similar typographical errors.*

*"...MDJ-03 site in Congo (5.98°S; 12.87°E)…"* **(In revision lines 361-362)**

**Comment 17:** 17. Greater attention should be devoted to the details of the figures in the manuscript. For example, The bolded font in Fig1 and Fig 3b. The labels of latitude and longitude in Figures 5-9 should be unified. Please standardize the style of all figures throughout the manuscript, particularly ensuring consistency in the map display, including the axes and other elements.

*Response:* *Thank you for these valuable suggestions. We have conducted an additional check to rectify these issues. In the revised manuscript, we reproduced all figures with unified format to eliminate formatting inconsistencies and typos. Specifically, we uniformed the font in each figure like Figs 1 and 3. We also revised the labels and scale of all figure to clearly display our result. Meanwhile, the same intervals of latitude and longitude and related labels were adopted for the all maps to ensuring consistency.*

[Figure]

**Figure 1.** *Tropical and subtropical broadleaved evergreen forests (TEFs) and in situ observation sites. The studied TEFs is determined as those labeled as evergreen broadleaf forest (EBF) from the MODIS land cover maps at a 0.05° spatial resolution. The red dots are in situ observation sites of $V_{c,max25}$.*

[Figure]

**Figure 3.** *Validations of simulated seasonal $V_{c,max25}$ for canopy-averaged leaves and young leaves with in situ observations. The green lines and green dots are the seasonal $V_{c,max25}$ of young leaf simulated from RTSIF derived GPP by the proposed method. The black line and red dots are the mean leaf age $V_{c,max25}$ values from the simulations and in situ observations, respectively. Simulated $V_{c,max25}$ denoted as the $V_{c,max25}$ of young leaf simulated from RTSIF-derived GPP by using the new proposed method. Mean $V_{c,max25}$ denoted as the mean leaves age $V_{c,max25}$.*

[Figure]

**Figure 5.** *The root mean square error (RMSE) and correlation coefficient (R) between the $V_{c,max25}$ of young leaves derived from RTSIF-derived GPP and that dissolved from GOSIF-derived GPP.*

[Figure]

**Figure 6.** *Comparison of sub-regions of the $V_{c,max25}$ of young leaf (a-c) with those of climatic factors classified by the K-means clustering analysis (d-f) analyzed by Chen et al. (2021), and those of the Leaf-age-dependent leaf area index seasonality product (Lad-LAI) (g-i) developed by Yang et al. (2023).*

[Figure]

*Figure 7. Spatial maps of the correlation coefficient (R) between the monthly simulated $V_{c,max25}$ and the Leaf-age-dependent leaf area index seasonality product (Lad-LAI) developed by Yang et al. (2023).*

[Figure]

*Figure 9. Spatial maps of correlation coefficient (R) between the SIF-simulated monthly $V_{c,max25}$ and climatic and phenological patterns. a, d and g are the spatial maps of correlation coefficient between $V_{c,max25}$ and vapor pressure deficit (VPD); b, e and h are the spatial maps of correlation coefficient between $V_{c,max25}$ and air temperature ($T_{mean}$); c, f and i are the spatial maps of correlation coefficient between $V_{c,max25}$ and downward shortwave solar radiation (SW)*

---

## Author Response (AR2)

**Responses to editor's comments point by point**

MS No.: essd-2025-64

Title: Remote sensing of young leaf photosynthetic capacity in tropical and subtropical evergreen broadleaved forests

Author(s): Xueqin Yang et al.

**Comment of Topical Editor:**

**Topical editor decision: Publish subject to minor revisions (review by editor)**

**Public justification (visible to the public if the article is accepted and published):**

Congratulations! Your manuscript has been accepted for publication, pending minor revisions. Please note that Table 1 has some formatting issues that need to be addressed.

*Response: Thanks for your valuable time in handling our manuscript and for your careful review. As per the journal's "Figures and Tables" submission guidelines, we have revised the formatting in **Table 1** accordingly.*

**Table 1. Data sources for mapping the $V_{c,max25}$ of young leaves across tropical and subtropical evergreen broadleaved forests**

| Data name (Abbr.) | Source | Usage | Spatial resolution | Temporal resolution | Temporal coverage |
|---|---|---|---|---|---|
| Temperature ($T_{mean}$) | ERA5-Land | Calculate the $K_C$, $K_0$, $\Gamma^*$, and $R_d$ for $A_n$ | 0.1°×0.1° | Monthly | 2001.1-2018.12 |
| Shortwave solar radiation (SW) | BESS | Calculate the $J_e$ for $A_n$ | 0.05°×0.05° | Monthly | 2001.1-2018.12 |
| Vapor pressure deficit (VPD) | ERA5-Land | Calculate the $C_i$ for $A_n$ | 0.1°×0.1° | Monthly | 2001.1-2018.12 |
| Sun induced chlorophyll fluorescence (RTSIF) | TROPOMI SIF | RTSIF-derived GPP | 0.05°×0.05° | Monthly | 2001.1-2018.12 |
| Gross primary production retrieved from OCO-2 Solar induced chlorophyll fluorescence (GOSIF) | GOSIF | GOSIF-derived GPP | 0.05°×0.05° | Monthly | 2001.1-2018.12 |
| Gross primary production from eddy covariance flux tower measurements (FLUXCOM) | FLUXCOM | FLUXCOM GPP | 0.5°×0.5° | Monthly | 2001.1-2013.12 |
| Leaf-age-dependent leaf area index (Lad-LAI) | Yang et al., 2023 | Dissolved $V_{c,max25}$ from GOSIF-derived GPP | 0.25°×0.25° | Monthly | 2001.1-2018.12 |

---

## Author Response (AR3)

**Responses to editor's remark point by point**

MS No.: essd-2025-64

Title: Remote sensing of young leaf photosynthetic capacity in tropical and subtropical evergreen broadleaved forests

Author(s): Xueqin Yang et al.

**Remarks from the preceding review file validation**

The ROR database lists the institution of the corresponding author but with a different city than given in the manuscript`s affiliation. Please clarify whether the ROR in the system "Sun Yat-sen University (Guangzhou, China)" is still correct.

***Response:*** *Thank you for your remark. The ROR entry "Sun Yat-sen University (Guangzhou, China)" is correct, as it reflects the official main campus location of the institution. The School of Atmospheric Sciences, to which the corresponding author is affiliated, is located in Zhuhai. Therefore, the manuscript affiliation correctly states Zhuhai as the city. Please let us know if any additional information is required.*